**Subject Category:**
Biology (whole organism)

evolution

myoanatomy, musculature, velvet worm, onychophora, tomography, three-dimensional reconstruction

**Author for correspondence:**
Ivo de Sena Oliveira
e-mail: ivo.oliveira@uni-kassel.de

[†]These authors shared first authorship.

# Functional morphology of a lobopod: case study of an onychophoran leg

Ivo de Sena Oliveira[1,2,†], Andreas Kumerics[1,†], Henry Jahn[1], Mark Müller[3], Franz Pfeiffer[3,4] and Georg Mayer[1]

[1]Department of Zoology, Institute of Biology, University of Kassel, Kassel, Germany
[2]Departamento de Zoologia, Instituto de Ciências Biológicas, Universidade Federal de Minas Gerais, Belo Horizonte, Brazil
[3]Chair of Biomedical Physics, Department of Physics and Munich School of Bioengineering, Technical University of Munich, Garching, Germany
[4]Department of Diagnostic and Interventional Radiology, Klinikum rechts der Isar, Technical University of Munich, 81675 München, Germany

IdSO, 0000-0002-8340-7660; GM, 0000-0003-0737-2440

Segmental, paired locomotory appendages are a characteristic feature of Panarthropoda—a diversified clade of moulting animals that includes onychophorans (velvet worms), tardigrades (water bears) and arthropods. While arthropods acquired a sclerotized exoskeleton and articulated limbs, onychophorans and tardigrades possess a soft body and unjointed limbs called lobopods, which they inherited from Cambrian lobopodians. To date, the origin and ancestral structure of the lobopods and their transformation into the jointed appendages are all poorly understood. We therefore combined high-resolution computed tomography with high-speed camera recordings to characterize the functional anatomy of a trunk lobopod from the onychophoran *Euperipatoides rowelli*. Three-dimensional reconstruction of the complete set of muscles and muscle fibres as well as non-muscular structures revealed the spatial relationship and relative volumes of the muscular, excretory, circulatory and nervous systems within the leg. Locomotory movements of individual lobopods of *E. rowelli* proved far more diverse than previously thought and might be governed by a complex interplay of 15 muscles, including one promotor, one remotor, one levator, one retractor, two depressors, two rotators, one flexor and two constrictors as well as muscles for stabilization and haemolymph control. We discuss the implications of our findings for understanding the evolution of locomotion in panarthropods.

# 1. Introduction

Panarthropoda comprises a diverse animal clade characterized, among other features, by segmental, paired locomotory appendages [1–4]. Extant panarthropods are classified into three major subclades including Onychophora (velvet worms), Tardigrada (water bears) and Arthropoda (spiders, centipedes, crustaceans, insects and allies), the latter being the most species-rich animal taxon [5–8]. The evolutionary origin of panarthropods, however, is very old (approx. 520 Ma) and deeply embedded in the Cambrian explosion [5], more specifically in a paraphyletic assemblage of extinct, marine animals commonly known as lobopodians (Greek *lobos* = lobe; *podos* = foot), in reference to their characteristic, unjointed limbs called lobopods (=lobopodia; sing. lobopodium) [6,7,9–11]. Interestingly, similar unjointed locomotory limbs are still found in the present-day onychophorans and tardigrades, while this ancestral type of appendages has been transformed into articulated limbs (=arthropodia; sing. arthropodium) in the arthropod lineage along with the evolution of a sclerotized exoskeleton [5,8,12,13]. The homology of the two major types of panarthropod limbs is supported by a similar proximodistal expression pattern of the leg gap genes *homothorax*, *extradenticle*, *dachshund* and *Distal-less* in onychophorans and arthropods [14].

To date, only little is known about the evolution of unjointed limbs in the panarthropod ancestor, and their transformation into the articulated limbs—a key innovation for the successful radiation and diversification of arthropods [15]—is poorly understood. Studies of evolutionary functional morphology suggest that this transformation might have been accompanied by an increase in limb length observed during evolution of lobopodians [5,16,17]. According to this scenario, longer limbs required a more precise regulatory mechanism for movement; thus, the role of extrinsic muscles in limb control may have expanded, and sclerotized structures may have evolved on the limbs and body wall to stabilize the muscle attachment sites [5]. Once these attachment sites were in place, they enlarged progressively until they were finally able to shift function from stabilization to protection, and to form articulations between them [5]. At the same time, a hydrostatic skeleton was gradually replaced by the lever-style muscular system observed in extant arthropods [5,17]. Although plausible, this hypothesis is difficult to test [5], since the incomplete preservation of most fossil lobopodians found to date and in particular their soft tissues including the musculature [3,17,18] precludes verification of the proposed series of morphological transformations.

Inferring the early origin of walking limbs in panarthropods, however, is problematic. Since fossil data lack resolution for assessing important anatomical details of limbs in lobopodians (e.g. number, arrangement and attachment sites of leg muscles), it is neither possible to assert how legs in the last common ancestor of Panarthropoda may have looked, nor to reconstruct the evolutionary events that may have led to the acquisition of this morphological trait [5]. Alternatively, data obtained from unjointed limbs of extant panarthropod taxa have been used to infer the evolution and putative functional morphology of the ancestral lobopodium [5,19]. Among panarthropods with unjointed limbs, comparison with tardigrades is problematic because these animals underwent miniaturization, due to which several morphological features, such as the heart and the circulatory system, have been either lost or strongly reduced (e.g. each tardigrade muscle consists of a single cell) [5,20–22]. Onychophorans, on the other hand, are exclusively terrestrial but more similar in morphology and body size to the Cambrian lobopodians than any other extant panarthropod group [23], thus being commonly used as a model for understanding the evolution of limbs in Panarthropoda [3,5,17,19,24–28].

The onychophoran lobopods (also called oncopods) resemble those of Cambrian lobopodians in several respects, including their fleshy and unjointed nature, the overall shape, the presence of distal claws and the existence of ventral pads functioning as walking soles [9,23,29,30]. These similarities suggest that the functional morphology of walking limbs in Cambrian lobopodians may have been indeed comparable to that of extant onychophorans [5,19]. However, the onychophoran leg itself has not yet been explored sufficiently and many aspects of its functional anatomy remain unclear. The few studies of the onychophoran lobopods mainly relied on traditional morphological techniques such as histological and semi-thin sections [24,27,28,31–33], which are prone to reconstruction artefacts and hinder proper interpretation of complex systems such as lobopodial musculature. To our knowledge, three-dimensional (3D) reconstruction of the complete set of muscles and muscle fibres associated with the onychophoran leg has not been carried out, nor have the movements of individual onychophoran legs been well characterized, with only a few studies briefly describing the coordination of legs and different gait patterns [27,34]. Consequently, the number, arrangement and putative function of individual muscles have been described inconsistently in the literature (summarized in table 1), thus proposing contradictory scenarios for the operational principles of the onychophoran leg, especially concerning the

**Table 1.** Onychophoran leg muscles revealed in the present and previous comprehensive studies on myoanatomy. Asterisks indicate muscles with uncertain identity.

| muscle | present study *Euperipatoides rowelli* (Peripatopsidae) | Snodgrass [28] *Peripatoides novaezealandiae* (Peripatopsidae) | Pflugfelder [33] *Paraperipatus amboinensis* (Peripatopsidae) | Manton [27,35] *Peripatopsis* spp. (Peripatopsidae) | Birket-Smith [31] *Peripatus dominicae* (Peripatidae) | Hoyle & Williams [24] *Peripatus dominicae* (Peripatidae) | comments |
|---|---|---|---|---|---|---|---|
| #1 | leg levator | dorsal remotor of the leg | (1) oblique and (2) transverse trunk musculature | dorsal remotor of the leg | dorsal remotor of the lobopod | remotor | Pflugfelder [33], followed by Manton [27,35], recognized the muscles #1–#4 based on their extrinsic parts, whereas their intrinsic parts were described together as a 'thick peripheral layer of intrinsic leg fibres' (='peripheral muscles of basal part of leg' by Snodgrass [36]; 'superficial longitudinal muscle' by Manton [27,35]). |
| #2 | leg depressor | ventral promotor of the leg | musculature | ventral promotor of the leg | ventral promotor of the lobopod | anterior depressor | |
| #3 | leg promotor | dorsal promotor of the leg | | dorsal promotor of the leg | dorsal promotor of the lobopod | promotor | |
| #4 | leg remotor | ventral remotor of the leg | | ventral remotor of the leg | ventral remotor of the lobopod | posterior depressor | |
| #5 | anterior leg rotator | — | — | — | — | — | |
| #6 | posterior leg rotator | — | — | — | — | — | |
| #7 | anteroproximal leg muscle | transverse muscle of the leg base | — | transverse muscle of the leg base | anterior retractor of the nephropore | — | |
| #8 | posteroproximal leg muscle | — | — | — | posterior retractor of the nephropore | — | |
| #9 | leg flexor | — | — | — | — | — | |
| #10 | anteroposterior septal muscle | anteroposterior septal muscle | — | transverse muscle | anteroposterior lobopod muscle sheet | septal muscle | |

(*Continued.*)

**Table 1.** (*Continued.*)

| muscle | present study<br>*Euperipatoides rowelli*<br>(Peripatopsidae) | Snodgrass [28]<br>*Peripatoides novaezealandiae*<br>(Peripatopsidae) | Pflugfelder [33]<br>*Paraperipatus amboinensis*<br>(Peripatopsidae) | Manton [27,35]<br>*Peripatopsis* spp.<br>(Peripatopsidae) | Birket-Smith [31]<br>*Peripatus dominicae*<br>(Peripatidae) | Hoyle & Williams [24]<br>*Peripatus dominicae*<br>(Peripatidae) | comments |
|---|---|---|---|---|---|---|---|
| #11 | dorsoventral septal muscle | — | — | — | transverse lobopod muscle sheet | — | the dorsoventral septal muscle (#11) was identified as muscle '14' by Birket-Smith [31] in reference to Snodgrass [28], although the latter used '14' for a muscle that corresponds to the anteroproximal leg muscle (#7) in our study instead |
| #12 | claw retractor | two-branched retractor of claws | levator of claws | levator muscle | lifter of the claws | retractor of the claw | the 'two branches' of the claw retractor described by Snodgrass [28], namely muscles '19a' and '19b', correspond to the single muscle #12 in the present study |
| #13 | foot depressor | flexor of distal leg rings | flexor muscle of penultimate foot segment | depressor muscle | flexor (protruder) muscle of the foot pads | — | this muscle was recently identified as 'foot retractor muscle' by Oliveira & Mayer [37] and as a 'protractor' of the foot by Müller et al. [38] |
| #14 | bridge constrictor | — | — | — | — | — | |
| #15 | foot constrictor | circular muscles of foot | — | — | protractor muscle of the claws | small circular fibres (…) of the foot | |
| * | — | flexor muscle of leg | — | — | flexor muscle of the lobopod | — | this muscle possibly represents a subset of muscle fibres belonging to the leg promotor (#3) of the present study |

(*Continued.*)

**5**

**Table 1.** (*Continued.*)

| muscle | present study *Euperipatoides rowelli* (Peripatopsidae) | Snodgrass [28] *Peripatoides novaezealandiae* (Peripatopsidae) | Pflugfelder [33] *Paraperipatus amboinensis* (Peripatopsidae) | Manton [27,35] *Peripatopsis* spp. (Peripatopsidae) | Birket-Smith [31] *Peripatus dominicae* (Peripatidae) | Hoyle & Williams [24] *Peripatus dominicae* (Peripatidae) | comments |
|---|---|---|---|---|---|---|---|
| * | — | — | — | (superficial) oblique muscles | — | — | — |
| * | — | — | (1) adductor and (2) remotor of the claws | — | depressor of the claws | — | — |
| * | — | — | — | circular muscles in the limb wall | external circular muscle | (...) circular muscle of the body wall running into the leg | — |
| * | — | — | — | 14a | — | — | Manton [27,35] neither named nor further characterized these muscles |
| * | — | — | — | 14b | — | — | |
| * | — | — | — | — | retractor muscle of the lobopod | — | this muscle possibly represents a subset of muscle fibres belonging to the leg levator (#1) in the present study |
| * | — | — | — | — | nephropore protractor muscle | — | — |
| * | — | — | — | — | ribbon-shaped lobopod muscle | — | — |

(*Continued.*)

**Table 1.** (*Continued.*)

| muscle | present study *Euperipatoides rowelli* (Peripatopsidae) | Snodgrass [28] *Peripatoides novaezealandiae* (Peripatopsidae) | Pflugfelder [33] *Paraperipatus amboinensis* (Peripatopsidae) | Manton [27,35] *Peripatopsis* spp. (Peripatopsidae) | Birket-Smith [31] *Peripatus dominicae* (Peripatidae) | Hoyle & Williams [24] *Peripatus dominicae* (Peripatidae) | comments |
|---|---|---|---|---|---|---|---|
| * | — | — | — | — | tensor muscle of the horizontal septum | — | — |
| * | — | — | — | — | internal, caudal sphincter muscle of the lobopod | — | — |
| * | — | — | — | — | retractor muscle of the eversible sac | — | Birket-Smith [31] used the term 'eversible sac' for the so-called coxal vesicle, which occurs on the ventral surface of the leg (e.g. [39]), rather than for the eversible dorsal sac associated with the foot (*sensu* [37]) |
| * | — | — | — | — | frontal sphincter muscle of the lobopod | — | — |
| * | — | — | — | — | external, caudal sphincter muscle of the lobopod | — | — |
| * | — | — | — | — | — | levator | — |
| number of identified muscles | 15 | 10 | 6 | 12 | 22 | 9 | |

spatial and functional relationships between the muscles and the role and regulation of the hydrostatic system within the leg [24,27,28,31,33].

To clarify the myoanatomy and to better understand the operational principles of a lobopod, we investigated the mid-trunk legs of the velvet worm *Euperipatoides rowelli* by combining high-resolution X-ray nano- and micro-computed tomography data with high-speed camera recordings. While nano- and micro-computed tomography allows resolutions down to 400 nm, providing enough detail to distinguish single bundles of muscle fibres [38,40,41], high-speed camera recordings enable an analysis of movements of individual legs. On the basis of the new data, we provide a 3D reconstruction of a complete onychophoran leg including its muscular, nervous, excretory and circulatory systems as well as cuticular structures such as the skin, claws and foot apodeme. These reconstructions reveal details of the spatial relationships and relative volumes of structures and cavities (=lacunar system) inside the leg. Analyses of high-speed camera footage shed light on the operation of the onychophoran leg during forward movement and show that the onychophoran locomotion is more complex and the locomotory movements more diverse than previously thought. In addition, we identified possible sources of previous inconsistencies and discuss the evolutionary implications of our findings in terms of evolution of locomotion in panarthropods.

# 2. Material and methods

## 2.1. Specimens

Specimens of the onychophoran species *Euperipatoides rowelli* Reid, 1996 (Peripatopsidae) were collected in the Tallaganda State Forest (New South Wales, Australia; 35°26′ S, 149°33′ E, 954 m) and kept in culture under controlled conditions as described elsewhere [42]. Selected specimens were analysed *in vivo* or anaesthetized with chloroform vapour and prepared further for different analyses as described below. Animals were collected under the permit numbers SL100159/2011 and SL101720/2016 issued by the NSW National Parks & Wildlife Service and exported under the permit numbers WT2012-8163 and PWS2016-AU-001023 obtained from the Department of Sustainability, Environment, Water, Population and Communities.

## 2.2. Micro-computed tomography

The synchrotron radiation-based micro-computed tomography (SRμCT) data were obtained from an adult female specimen of *E. rowelli* fixed overnight in paraformaldehyde (PFA; 4% in phosphate-buffered saline (PBS), 0.1 M, pH 7.4), contrasted in osmium tetroxide ($OsO_4$, 2% in water overnight, Science Services GmbH, Munich, Germany), dehydrated in an ascending ethanol series, dried in a CPD 030 critical point dryer (BAL-TEC AG, Balzers, Liechtenstein) and scanned at the beamline P05 of the storage ring PETRA III (Deutsches Elektronen-Synchrotron—DESY, Hamburg, Germany) as described by Jahn *et al.* [41].

## 2.3. Nano-computed tomography

The X-ray nano-computed tomography (nanoCT) dataset analysed herein was previously generated by Müller *et al.* [38] and consisted of 1200 z-stack images of a left leg from the mid-body of a female of *E. rowelli*. For nanoCT analyses, a newborn specimen was fixed and stored in 4% formaldehyde in PBS for several weeks. After several washes in PBS, the specimen was cut into pieces each containing a single leg with a small part of the body wall. Thereafter, the pieces were contrasted in 1% $OsO_4$ overnight, dehydrated in an ethanol series, critical point dried and mounted onto standardized nanoCT sample holders. Acquisition, processing and reconstruction of nanoCT data were carried out as described by Müller *et al.* [38].

## 2.4. Three-dimensional reconstruction and relative volume estimates

Different tissue types and structures were initially recognized in the dataset based on their specific grey values, texture and orientation within the leg. Segmentation of leg tissues and structures was carried out manually using the software Amira 6.0.1 (FEI Visualisation Sciences Group, Burlington, MA, USA). Three-dimensional volume renderings of segmented structures were generated using the software

VGStudio Max 3.0 (Volume Graphics GmbH, Heidelberg, Germany). Estimates of the relative volumes were obtained by assessing the number of voxels in each segmented structure in relation to the total number of voxels of the entire leg. For that, regions of interest (ROI) were created in VGStudio to initially separate leg and body regions. Structures (or parts of them) lying within the leg were exported individually as image stacks, imported into the freeware FIJI v. 1.52j [43] and hollow regions were filled using the Process > Binary > Fill Holes option. The filled structures were imported into VGStudio to obtain their respective numbers of voxels.

## 2.5. Scanning electron microscopy

Adult female specimens of *E. rowelli*, as well as moulted skins preserved in 70% ethanol, were cut into small pieces containing four leg pairs. Body parts were transferred to distilled water through a descending ethanol series, fixed overnight in 4% PFA and washed several times in PBS. Selected pieces were subsequently embedded in albumin–gelatin medium (Sigma-Aldrich, St Louis, USA; 3.75 g in 10 ml distilled water and 0.5 g in 2.5 ml distilled water, respectively), cooled down at 4°C for 4 h, postfixed in 10% PFA overnight and sectioned using a vibratome (Micron HM 650 V; Thermo Scientific, Walldorf, Germany). Body parts, skin pieces and sections were dehydrated in an ascending ethanol series, dried in a critical point dryer (CPD 030, BAL-TEC AG), mounted onto standardized stubs, coated with gold-palladium in a Polaron SC 7640 sputter coater (VG Microtech, East Grinstead, West Sussex, UK) and imaged in a Hitachi S4000 field emission scanning electron microscope (Hitachi High-Technologies, Europe GmbH, Düsseldorf, Germany) as described previously [44,45].

## 2.6. High-speed camera analyses

The locomotion of living specimens of *E. rowelli* was recorded using a high-speed camera (Phantom Miro LC320S; Vision Research Phantom, Wayne, NJ, USA) equipped with a macro lens (105 mm) at 500 fps. Walking animals were recorded on a piece of tree bark moistened with distilled water to simulate the substrate of a typical onychophoran habitat. Motion sequences were analysed using the software Adobe Premiere Pro CS5.5 (Adobe Systems Incorporated, San Jose, CA, USA).

## 2.7. Image processing and panel design

Final image and movie processing was performed with Adobe Photoshop CS5.1 and Adobe Premiere Pro CS5.5. Illustrations and panels were designed with Adobe Illustrator CS5.1 (Adobe Systems Incorporated, San Jose, CA, USA).

# 3. Results

## 3.1. Anatomy of the lobopod excluding the musculature

The leg (le) of *E. rowelli* exhibits an externally annulated cuticle with transverse rings of dermal papillae (dp) (figure 1a). The leg bears a distal foot equipped with a pair of sickle-shaped claws (cl) and three distal foot papillae (fp) (one anterior, one dorsal and one posterior). The foot is connected with the remaining portion of the leg via a triangular ventral bridge (br; figure 1a). The bridge shows a deep, longitudinal median furrow, which corresponds to the foot apodeme (ap; figure 1a–f). The paired claws appear either protracted (figure 2a) or retracted into the foot (figure 2b), depending on their condition during the fixation. In the fully retracted state, only the smooth tips of the claws remain outside the foot (figure 2c). When protracted, a conspicuous dorsal sac (es) becomes externally visible at the basis of the claws, while the three distal foot papillae become erected, with their sensory bristles pointing in the dorsal, anterior and posterior directions (arrows in figure 2a). When the claws are retracted, the eversible sac is inverted back into the foot and the distal foot papillae bend towards the claws, so that their sensory bristles point in the distal direction (arrow in figure 2b). On the ventrodistal portion of the leg, three arch-shaped spinous pads (sp) occur next to the bridge (figure 1a). While the first and the second pads are adjacent to each other, the second and the third spinous pads are separated by a narrow, spineless integumentary fold (if; arrow in figure 1a; electronic supplementary material, figure S1a,b; electronic supplementary material, Movie S1). At the

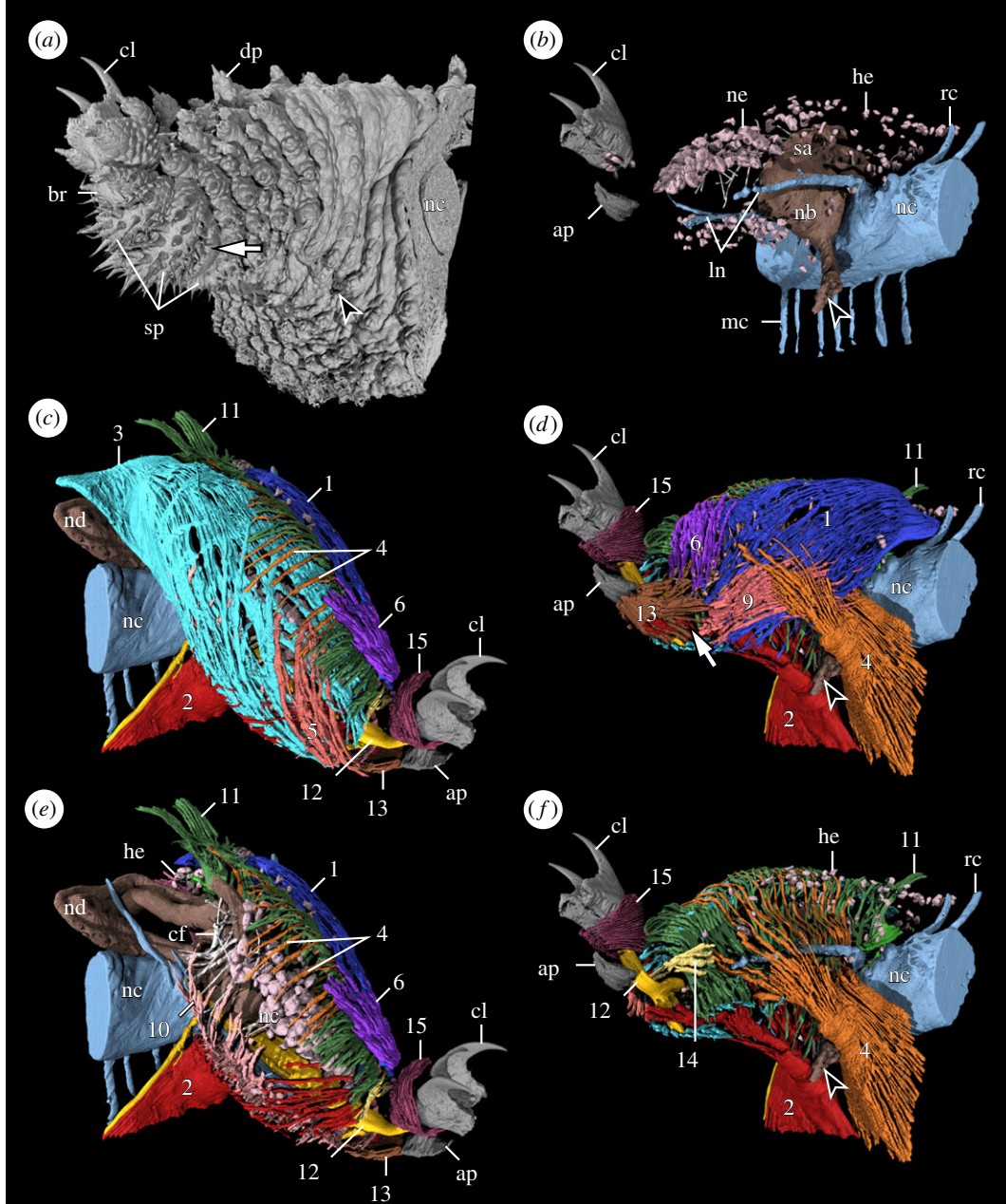

**Figure 1.** External and internal anatomy of the lobopod in *E. rowelli*. Volume rendering based on nanoCT data from the left mid-trunk leg in posterior (*a,b,d,f*) and anterior views (*c,e*). Dorsal is up in all images. Arrowheads demarcate the position of nephridial opening; arrows indicate spineless integumentary fold located between second and third spinous pads. (*a*) External structures, (*b*) internal non-muscular structures and (*c–f*) myoanatomy. Individual muscles are highlighted in different colours and numbered as in main text (summarized in table 1). Note that (*e,f*) illustrate the same perspective as (*c,d*) except that some outer muscles (#3, #5 in (*e*), and #1, #6, #9, #10; #13 in (*f*)) were excluded to better visualize internal muscles. ap, foot apodeme; br, foot bridge; cf, large collagen fibres; cl, claw; dp, dermal papilla; he, haemocyte; ln, leg nerves; mc, median commissure; nb, nephridial bladder; nc, nerve cord; nd, nephridial duct; ne, nephrocyte; rc, ring commissure; sa, sacculus; sp, spinous pads.

ventral leg basis, the nephridial opening (no; arrowhead in figure 1*a,b,d,f*) appears as an inconspicuous longitudinal slit between the surrounding dermal papillae.

The internal space of the leg is largely occupied by an elaborate lacunar system composed by channels and cavities, which are filled by haemolymph (=blood) and account for nearly half of the leg volume in a fixed specimen (table 2; electronic supplementary material, Movie S2). These include: (i) numerous transverse channels beneath the transverse rings of dermal papilla (tc; figure 3*a–c,e*); (ii) the four major compartments I–IV separated by the anteroposterior and dorsoventral septal muscles

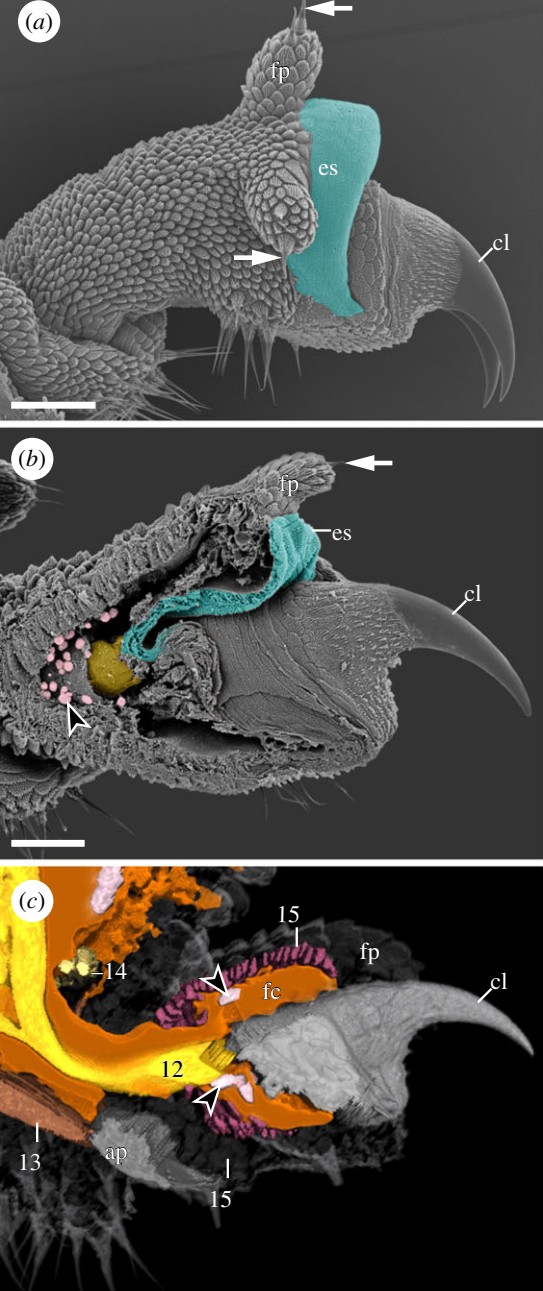

**Figure 2.** External and internal anatomy of the foot in *E. rowelli*. Scanning electron micrographs (*a,b*) and volume rendering of nanoCT data (*c*). Dorsal is up in all images. Arrows in (*a,b*) indicate sensory bristles of distal foot papillae. (*a*) Foot with protracted claws. Note everted dorsal sac (pseudocoloured turquoise). (*b*) Sagittal section of the foot with retracted claws. Note contracted eversible dorsal sac and distal foot papilla bent distally. Also note attachment of claw retractor muscle (pseudocoloured yellow) at claw base and presence of haemocytes (pseudocoloured pink) in foot cavity. (*c*) Virtual sagittal section of the foot. The leg and foot cavities illustrated in light-brown/orange; muscles colour-coded as in figure 1 and numbered as in main text (summarized in table 1). Arrowheads point to haemocytes located in foot cavity. ap, foot apodeme; cl, claw; es, eversible dorsal sac; fc, foot cavity; fp, distal foot papilla. Scale bars: 30 μm in (*a*) and 50 μm in (*b*).

(figure 3*c*–*e*); and (iii) the foot cavity (fc) surrounding the claw bases and extending into the dorsal eversible sac (figures 2*c* and 3*a*–*c*,*e*; electronic supplementary material, figure S2a). All these channels and cavities are confluent with each other and connected to the main body cavity via a narrow channel dorsolateral to the nerve cord (arrow in figure 3*e*). Among the four major compartments of the leg cavity (lc), the two dorsal ones (I and II) are the largest (figure 3*c*–*e*).

**Table 2.** Relative volumes of structures within the leg of *E. rowelli*. For volume estimates of individual muscles, only their 'intrinsic' portions were taken into account.

| structure, cavity, cell type or tissue | | total number of voxels | relative volume (%) |
|---|---|---|---|
| entire leg | | 178 999 899 | 100.00 |
| leg cavity | | 81 829 369 | 45.71 |
| *muscles* | | | |
| #1 | leg levator | 4 153 183 | 2.32 |
| #2 | leg depressor | 3 933 669 | 2.20 |
| #3 | leg promotor | 7 450 512 | 4.16 |
| #4 | leg remotor | 2 072 857 | 1.16 |
| #5 | anterior leg rotator | 256 237 | 0.14 |
| #6 | posterior leg rotator | 284 769 | 0.16 |
| #7 | anteroproximal leg muscle | 259 261 | 0.14 |
| #8 | posteroproximal leg muscle | 210 226 | 0.12 |
| #9 | leg flexor | 916 491 | 0.51 |
| #10 | anteroposterior septal muscle | 3 059 215 | 1.71 |
| #11 | dorsoventral septal muscle | 1 811 738 | 1.01 |
| #12 | claw retractor | 1 497 049 | 0.84 |
| #13 | foot depressor | 266 166 | 0.15 |
| #14 | bridge constrictor | 59 578 | 0.03 |
| #15 | foot constrictor | 227 010 | 0.13 |
| total | | | 14.78 |
| *non-muscular structures* | | | |
| nervous system | | 970 481 | 0.54 |
| excretory system | | 10 676 174 | 5.96 |
| haemo- and nephrocytes | | 1 956 540 | 1.09 |
| large collagen fibres | | 81 718 | 0.05 |
| claws | | 958 206 | 0.54 |
| foot apodeme | | 117 422 | 0.07 |
| cuticle | | 10 365 894 | 5.79 |
| other cells and tissues[a] | | 45 586 134 | 25.47 |
| total | | | 39.51 |

[a]Structures not segmented herein, including epidermal cells, peripheral sensory neurons, extracellular matrix and connective tissue.

Two types of blood cells, including haemo- (he) and nephrocytes (ne), are found within the haemal spaces of the leg. The haemocytes appear as small, individual cells that occur randomly in the entire haemolymph system of the leg, including the foot (figure 1*b,e,f*; arrowheads in figure 2*b,c*; electronic supplementary material, figure S2b and Movie S1). The nephrocytes, by contrast, are relatively larger and aggregated into clusters of three to six cells. The clusters of nephrocytes are mainly localized in the anterodorsal (I) blood compartment (electronic supplementary material, figure S2b), where they are not only attached to each other but also to the surrounding muscles and other tissues via a mesh of large collagen fibres (cf; figures 1*b,e* and 3*e,f*).

Additional non-muscular structures of the lobopod include elements of the nervous and excretory systems (electronic supplementary material, figure S3). The onychophoran trunk (tr) exhibits a pair of ventrolateral nerve cords (nc) that are linked with each other by a series of ring (rc) and median (mc) commissures (figure 1*b*; see [46] for details). The ring commissures are absent in the leg-bearing regions, which instead show prominent pairs of segmental nerves, the anterior and posterior leg

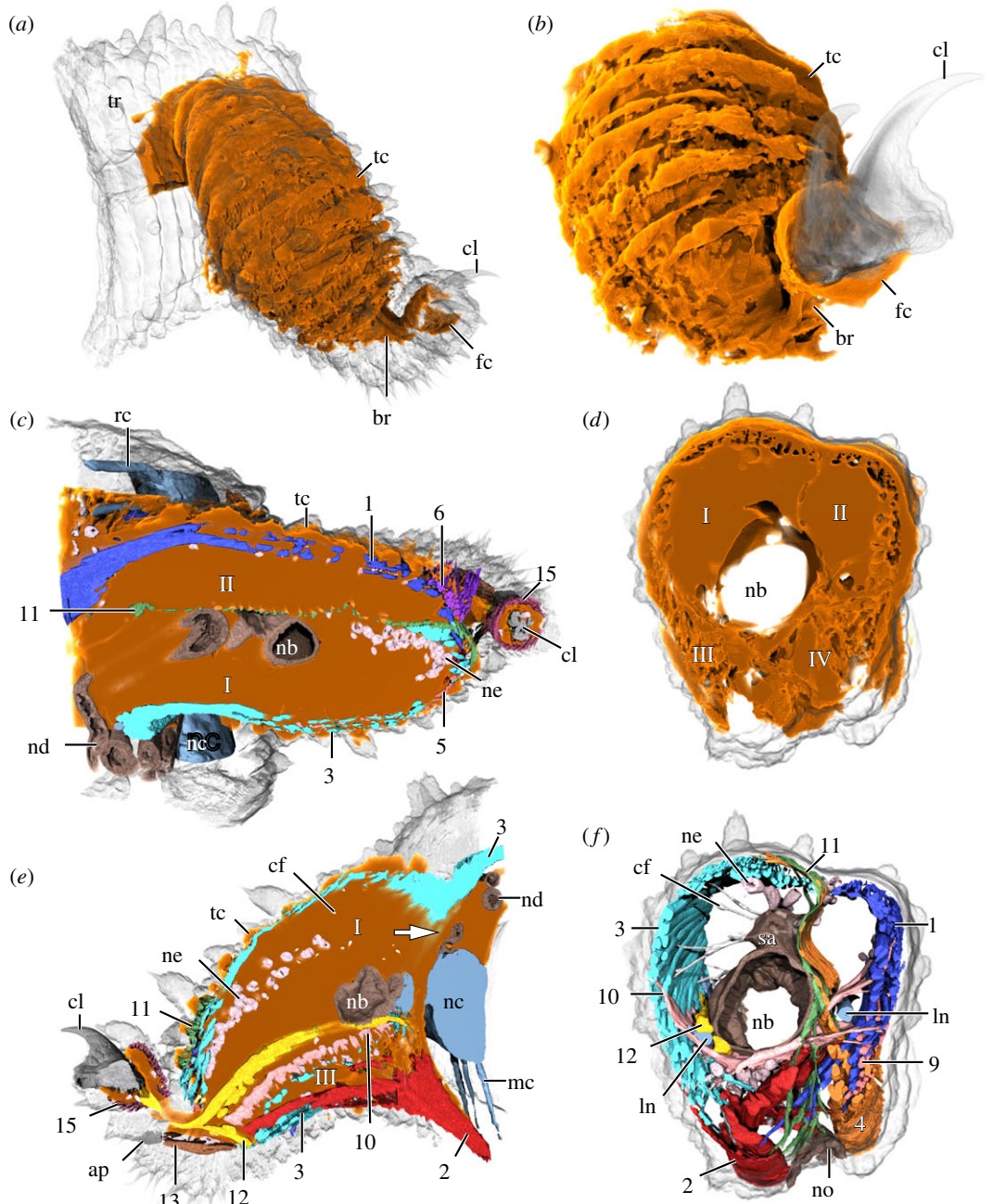

**Figure 3.** Lacunar system of the lobopod in *E. rowelli*. Volume rendering of nanoCT data from the left mid-trunk leg showing space occupied by haemolymph. Dorsal is up in (*a,b,d–f*); posterior is up in (*c*). Compartments of leg cavity are numbered I–IV. (*a,b*) Overview of the lacunar system in leg and foot. Note that transverse, haemal ring channels correspond to external rings of dermal papillae (semi-transparent in (*a*)). (*c–e*) Horizontal (*c*), cross (*d*) and sagittal (*e*) virtual sections of the leg. The arrow in (*e*) points to narrow channel connecting cavities of the leg and trunk. Muscles are numbered as in main text (summarized in table 1). Note that anteroposterior and dorsoventral septal muscles (#10, #11) subdivide leg cavity into four compartments (I–IV in (*d*)), with two dorsal compartments (I and II) being largest. (*f*) The same virtual section and perspective as in (*d*) showing arrangement of internal structures to the exclusion of leg cavity. ap, foot apodeme; br, foot bridge; cf, large collagen fibres; cl, claw; fc, foot cavity; ln, leg nerve; mc, median commissure; nb, nephridial bladder; nc, nerve cord; nd, nephridial duct; ne, nephrocytes; no, nephridial opening; sa, sacculus; rc, ring commissure; tc, transverse haemal ring channels; tr, trunk.

nerves (ln); these nerves arise from the dorsolateral portion of the nerve cord and extend into each leg (figure 1*b*). The two leg nerves give rise to numerous fibres (not reconstructed herein) supplying the musculature and the epidermis. The excretory organ of the leg (=nephridium) is mainly located in the proximal part of the leg and consists of the proximal sacculus (sa) connected via a convoluted

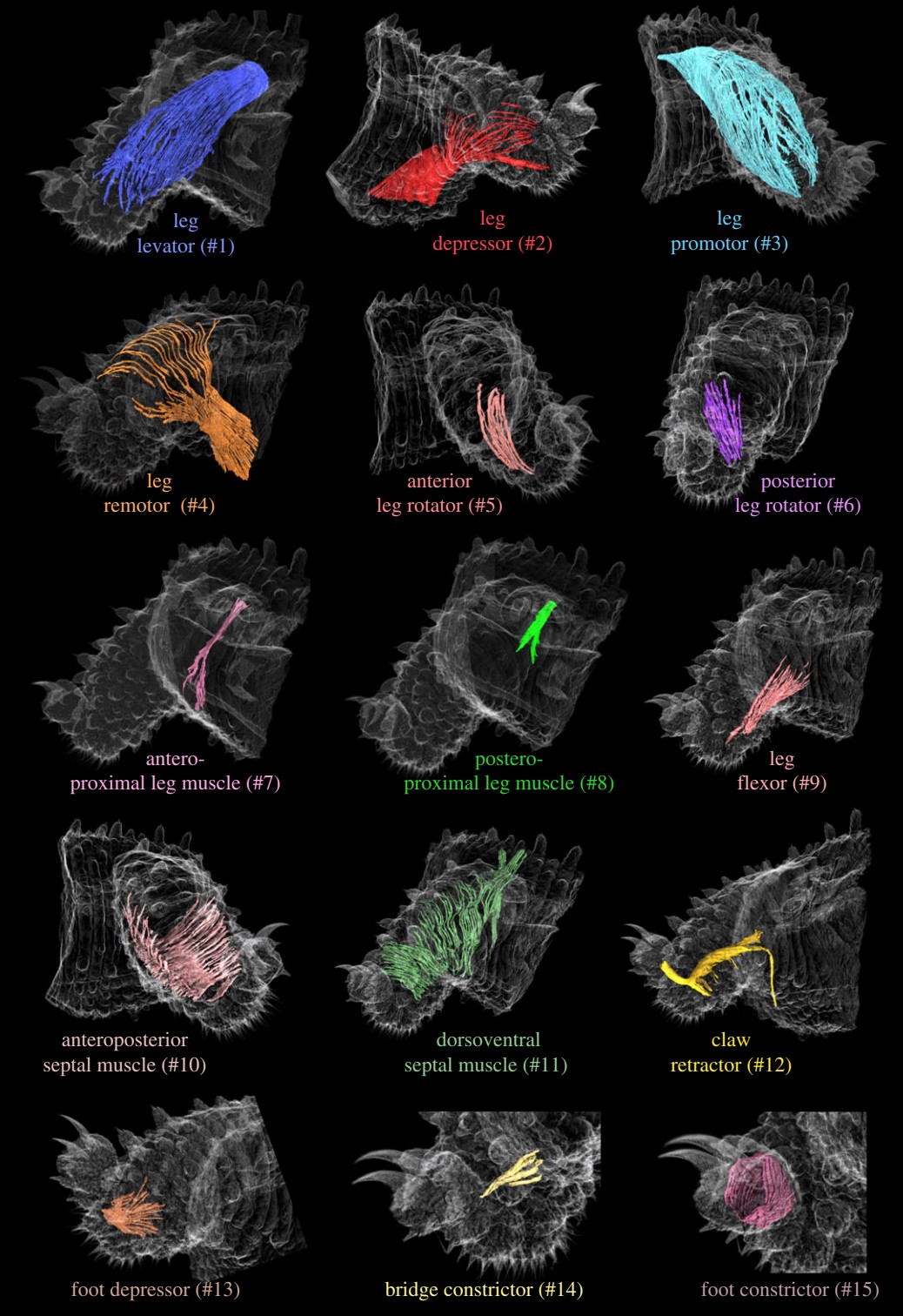

**Figure 4.** Overview of individual muscles associated with the lobopod of *E. rowelli*. Volume rendering of nanoCT data from the left mid-trunk leg illustrating the shape and position of all 15 lobopodial muscles. Dorsal is up in all images. Muscles are numbered as in main text (summarized in table 1). Body surface is semi-transparent; 'extrinsic' portions of muscles #1–#4 and #8 are not shown.

nephridial duct (nd) to the distal bladder (nb), which tapers into a short excretory duct associated with the nephridial opening (arrowhead in figure 1*b*,*d*,*f* and figure 3*c*–*f*; electronic supplementary material, figure S3). The nephridial duct extends from the sacculus into the trunk, then loops back into the leg cavity and finally joins the bladder (figure 1*b*,*e*).

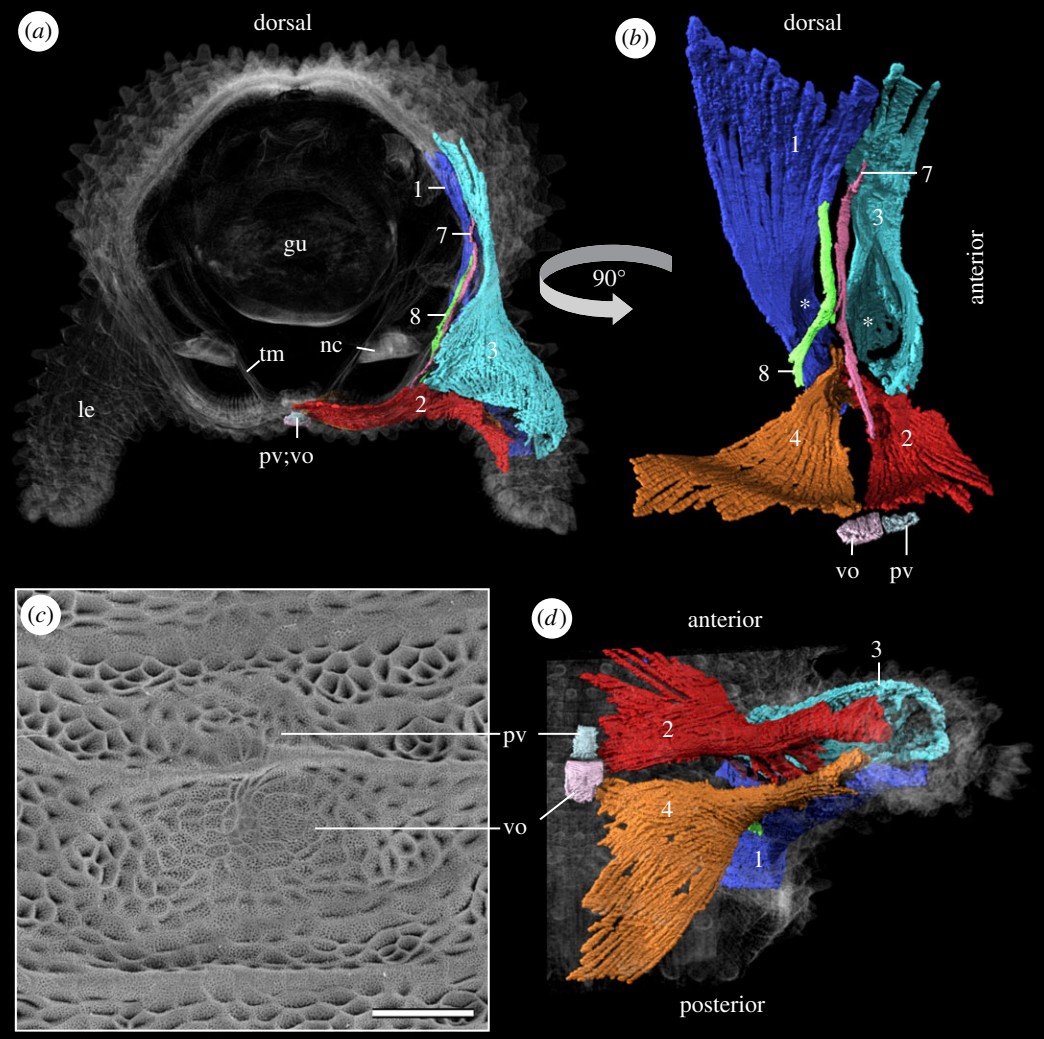

**Figure 5.** Lobopodial muscles extending into the trunk of *E. rowelli*. Volume rendering based on SRμCT data (*a,b,d*) and scanning electron micrograph (*c*). Muscles are numbered as in main text (summarized in table 1). (*a*) Virtual slice of the mid-body trunk segment. (*b*) Internal perspective of 'extrinsic' leg muscles viewed from the trunk. Note the median position of anteroproximal (#7) and posteroproximal leg muscles (#8) crossing the main channel (asterisks) linking cavities of the leg and trunk. (*c*) Imprints of ventral and preventral organs on the inner surface of moulted cuticle. (*d*) Ventral view of lobopod showing relationship of leg depressor (#2) and leg remotor (#4) with ventral and preventral organs. gu, gut; le, leg; nc, nerve cord; pv, preventral organ; tm, transverse musculature; vo, ventral organ. Scale bar: 50 μm in (*c*).

## 3.2. Myoanatomy of the lobopod

Segmentation and 3D reconstruction of individual bundles of muscle fibres revealed an intricate meshwork of 15 muscles within the lobopod of *E. rowelli* (electronic supplementary material, Movies S1–S4). In the following, these muscles are numbered consecutively (#1–#15) and named according to their position within the leg and/or presumed function (table 2 and figure 4).

### 3.2.1. Leg levator (#1)

This muscle forms a sheet, which embraces the posterodorsal (II) blood compartment (figures 1*d*, 3*c,f* and 4). Its proximal fibres project into the trunk and attach to the dorsolateral body wall, but the cuticle covering this region does not show any apodeme or apodeme-like structure (figure 5*a,b*; electronic supplementary material, figure S4). Relatively flat bundles of fibres belonging to the leg levator fan out in both dorsal ('extrinsic' portion) and ventral directions ('intrinsic' portion) and are associated mainly with the posterior leg region (figures 4 and 5*b*). When followed dorsoventrally, the muscle

undergoes a counterclockwise torsion of approximately 90° (figures 4 and 5*b*). Distally, most fibres of the leg levator attach to the posteroventral surface of the leg and the spineless integumentary fold. However, a small subset of ventralmost fibres crosses the leg anteriorly and attaches to the anteroventral epidermis (figure 3*f*); another small subset of anteriormost fibres runs along the posterodorsal border of the leg and attaches distally to the epidermal folds between the transverse rings of the integument (figure 1*d*).

### 3.2.2. Leg depressor (#2)

The leg depressor occupies the anteroventral portion of the leg (figures 3*e,f*, 4 and 5*a,b,d*). The proximal fibres of this muscle run into the trunk towards the ventral midline (figure 5*a,b,d*), where a large set of fibres attaches to epidermal cells that form the ventral (vo) and preventral (pv) organs—segmental, apodeme-like structures (figure 5*a–d*). Within the leg, the fibres of the leg depressor form a flat and dense muscle sheet that undergoes an approximately 180° torsion while crossing the anteroventral (III) blood compartment (figures 1*d,f*, 3*f*, 4 and 5*b,d*). Some fibres extend in the posteroventral leg region and attach next to the distal margin of the nephridial opening and further distal between the first and the second spinous pad (figures 3*e*, 4 and 5*d*). A small set of fibres fan out and attach to the anterodistal portion of the leg (figures 1*e* and 4).

### 3.2.3. Leg promotor (#3)

This is the most prominent leg muscle (table 2). It occupies most of the anterodorsal part of the leg and forms a sheet that covers the anterodorsal (I) blood compartment (figures 1*c*, 3*c–f*, 4 and 5*a,b,d*). The 'extrinsic' portion of the leg promotor is formed by fibres that attach along the dorsolateral body wall, not being associated with any type of apodeme-like structure (electronic supplementary material, figure S4). Near the leg basis, fibres belonging to the leg promotor become densely packed, forming a flat bundle that surrounds the anterodorsal blood compartment (figures 3*c,e,f*, 4 and 5*a,b*). A small set of anteriormost extrinsic fibres surrounds the anterior half of the haemolymph channel opening into the body cavity and attaches to the ventral leg basis (figure 5*a,b*), while the remaining fibres fan out into the leg (figures 1*c* and 4). The numerous fibres composing the 'intrinsic' portion of the leg promotor follow two distinct pathways: anterior fibres project distally along the anterior leg region and attach ventrally to the spineless integumentary fold and the second spinous pad (figures 1*c*, 3*f* and 4; electronic supplementary material, figure S5a,b); posterior fibres run dorsomedially (figures 1*c*, 3*c,e* and 4) and attach to the dorsodistal leg region (figure 4).

### 3.2.4. Leg remotor (#4)

This muscle occupies the ventroposterior portion of the leg. Proximally, its 'extrinsic' fibres extend to the ventral midline of the trunk (figure 5*b,d*), where a small set of anteriormost fibres attaches to the epidermal cells of the ventral organ (figure 5*b–d*), whereas their great majority fans out and attaches further posteriorly (figure 5*b,d*). When followed distally, the fibres of the leg remotor converge into a dense muscle that crosses the posteroventral (IV) blood compartment as it undergoes an approximately 90° rotation (figures 1*d,f*, 3*f*, 4 and 5*b,d*). The 'intrinsic' portion of the leg remotor is composed of a small number of ventralmost fibres projecting distally and attaching to the posteroventral leg region (figure 1*d*) and numerous bundles of sparsely arranged fibres, which run medially towards the dorsal leg region and attach anterodorsally (figures 1*d,e*, 3*f* and 4*)*.

### 3.2.5. Anterior leg rotator (#5)

This is a short 'intrinsic' muscle located distally in the anterior leg region (figures 1*c* and 4). Proximally, the parallel fibres of the anterior leg rotator embrace the anterodorsal (I) blood compartment, where they attach to the leg surface (figures 1*c*, 3*c* and 4). In its distal portion, fibres run ventrally and attach to the ridges of the spinous pads (figure 1*c*). This muscle is not associated with an apodeme-like structure.

### 3.2.6. Posterior leg rotator (#6)

The posterior leg rotator is an 'intrinsic' sheet-like muscle formed by a set of fibres surrounding the distal portion of the posterodorsal (II) blood compartment (figures 1*c–e*, 3*c* and 4). Ventrally, the fibres of the posterior leg rotator attach to the posterior edge of the spineless integumentary fold (arrow in figure 1*a,d*) as well as along the border of the second spinous pad. The proximal portion of this muscle attaches

medially to the dorsal surface of the leg (figures 1*d,e* and 4). This muscle is not associated with an apodeme-like structure.

### 3.2.7. Anteroproximal (#7) and posteroproximal (#8) leg muscles

These two muscles are formed by small sets of densely arranged fibres that dorsoventrally cross the main channel connecting the cavities of the leg and the trunk (figures 4 and 5*a,b*). The 'extrinsic' portion of these muscles extends to the dorsolateral body wall of the trunk, while the 'intrinsic' portion attaches to the ventral surface of the leg basis. More specifically, the ventral fibres of the anteroproximal muscle (#7) project anteriorly towards the leg depressor (#2; figure 5*b*), whereas those of the posteroproximal muscle (#8) arch posteriorly and attach further posteriorly (figure 5*b*). Neither of these two muscles are associated with the nephridium or apodeme-like structures (electronic supplementary material, figure S6a,b).

### 3.2.8. Leg flexor (#9)

The flexor is an 'intrinsic' muscle embracing the posteroventral (IV) blood compartment (figures 1*d* and 4). Distally, its fibres are densely packed and mainly attach to the ridge of the third spinous pad, with only a few of them connecting to the ridge of the spineless integumentary fold (figures 1*d* and 4). The fibres of the proximal part of the leg flexor fan out and attach to the posterior surface of the leg (figures 1*d* and 4). This muscle is not associated with an apodeme-like structure.

### 3.2.9. Anteroposterior septal muscle (#10)

This muscle is formed by numerous parallel, 'intrinsic' fibres that run along the anteroposterior axis of the leg and form a septum-like sheet, which subdivides the leg cavity into dorsal (I + II) and ventral (III + IV) blood compartments (figures 1*e*, 3*e,f* and 4). Bundles of fibres belonging to this muscle appear spaced anteriorly and posteriorly, i.e. near their attachment points to the anterior and posterior walls of the leg, whereas the fibres converge into a compact muscle medially (figure 4). The anteroposterior septal muscle extends nearly through the entire leg from the level of the nephropore to the level of the second spinous pad and borders ventrally the nephridial bladder and the two leg nerves (figures 1*e*, 3*e,f* and 4). It is not associated with an apodeme-like structure.

### 3.2.10. Dorsoventral septal muscle (#11)

This muscle appears as a sheet of more or less parallel fibres that are more compact in the distal leg region and span the leg dorsoventrally, thus subdividing its cavity medially into anterior (I + III) and posterior (II + IV) blood compartments (figures 1*c–f*, 3*c,f* and 4). The dorsoventral septal muscle extends from the level of the nerve cord (proximal limit) to the second spinous pad (distal limit) (figures 1*e,f*, 3*c* and 4). A small subset of proximal fibres projects dorsally into the trunk, where they are associated with the lateral body wall ('extrinsic' portion), and ventrally into the leg, where they attach to the wall of the leg basis next to the nephridial opening (figure 1*c–f*). However, most fibres belonging to this muscle are still located within the leg ('intrinsic' portion), with their attachment points distributed along the dorsal and ventral proximodistal midlines of the leg (figures 1*e,f*, 3*c* and 4). This muscle is not associated with an apodeme-like structure.

### 3.2.11. Claw retractor (#12)

The claw retractor is a dense, prominent muscle that runs along the ventral area of the anterodorsal (I) blood compartment and crosses the bridge into the foot (figures 1*e*, 3*e* and 4; electronic supplementary material, figure S5). Distally, the claw retractor connects to the dorsal rim of each claw (figures 2*c*, 3*e* and 4; electronic supplementary material, figure S7a), while proximally, this muscle fuses with the leg promotor (#3) and their fibres become indistinct from each other (electronic supplementary material, figure S7b). The claw retractor possibly attaches along the dorsolateral body wall together with the leg promotor (#3). Several bundles of fibres branch off along the retractor and project ventrally (electronic supplementary material, figure S7a,b). The two most prominent bundles include a distal branch, which attaches to the spineless integumentary fold, and a proximal 'extrinsic' branch, which projects into the trunk and attaches to the ventrolateral body wall (figures 3*e* and 4). Additional sets of fibres branch off this muscle in a helical pattern and attach to the ventral wall of the leg, thus giving the

claw retractor a twisted appearance (figure 4; electronic supplementary material, figure S7a,b). This muscle is not associated with an apodeme-like structure.

### 3.2.12. Foot depressor (#13)

This 'intrinsic' muscle is located ventrally in the distal portion of the leg (figures 1c–e, 2c, 3e and 4). It appears as a flat muscle and attaches distally to the foot apodeme located in the bridge (figures 1d, 2c and 3e). Its proximal fibres fan out and attach to the edges of the third spinous pad and the spineless integumentary fold (figure 1a,d).

### 3.2.13. Bridge constrictor (#14)

The bridge constrictor is the least prominent leg muscle. It is composed of a few 'intrinsic' fibres located dorsal to the bridge (figures 1f, 2c and 4). Its fibres are oriented anteroposteriorly and attach to the anterior and posterior walls of the leg. This muscle is not associated with an apodeme-like structure.

### 3.2.14. Foot constrictor (#15)

This muscle consists of more or less parallel, circular fibres located underneath the surface of the foot (figure 4; electronic supplementary material, figure S2a). The ring fibres run perpendicularly to the main axis of the foot and surround nearly the entire foot cavity including that of the retracted eversible dorsal sac (figures 1c–f, 2c, 3c,e and 4). Some rings appear incomplete ventrally, while others show a denser ventral than dorsal arrangement (electronic supplementary material, figure S2a).

## 3.3. Relative volumes of structures and cavities within the lobopod

Relative volume estimates (table 2) reveal that the haemolymph space occupies nearly half of the lobopod (45.71%), followed by the epidermis and connective tissue including extracellular matrix and large collagen fibres (25.47%), and muscular (14.78%) and non-muscular structures (14.04%). Among non-muscular structures, elements of the excretory system (5.96%) and the cuticle (5.79%) are the largest. The most voluminous muscle (taking only the 'intrinsic' muscular portions into account) is the leg promotor (#3; occupying 4.16%), while the least voluminous muscle is the bridge constrictor (#14; occupying only 0.03% of the total leg volume).

## 3.4. Locomotory movements of individual lobopods during forward walking

Specimens of E. rowelli are able to walk either forward or backward. They exhibit different types of gaits, depending on structure and properties of the substrate, experimental conditions and whether or not the specimen is stressed. During locomotion, even different body regions of the same specimen may show different gait patterns, in which the two lobopods of the same segment are moved either synchronously or out of phase (figure 6). High-speed camera recordings of walking specimens in lateral view demonstrate that the movements of individual lobopods during forward walking involve seven major operational modes: (i) levation or depression; (ii) promotion or remotion; (iii) stretching, contracting and/or arching; (iv) anterior and posterior rotation (up to approx. 180°); (v) fine rotation of the distal leg portion (bearing the spinous pads); (vi) foot levation or depression; and (vii) claw protraction or retraction (electronic supplementary material, Movie S5).

Considering an individual lobopod of E. rowelli walking forward on a piece of wood (simulating the substrate of a typical microhabitat), its movements can be characterized as follows (figure 6a–h, a'–h'; electronic supplementary material, Movie S5). At the starting position (figure 6a,a'), the leg is inclined at approximately 45° posteriorly, the spinous pads are in contact with the substrate, the foot is levated and the claws are retracted. During leg levation, the leg is lifted from the substrate maintaining its initial 45° inclination relative to the body (figure 6b,b'). Thereafter, the leg is promoted above the substrate, rotates at approximately 90° and passes from a posterior to an anterior inclination of approximately 45° (figure 6b–d,b'–d'). During the levation and promotion of the leg, the foot remains levated. The leg depression begins when the spinous pads are facing anteriorly (figure 6d,d') and the leg is then pushed towards the ground until the posterior margins of the spinous pads touch the substrate (figure 6e,f,e',f'). At this point, the foot is depressed and the claws are protracted, hooking onto the substrate. The protraction of the claws is accompanied by the eversion of the dorsal sac

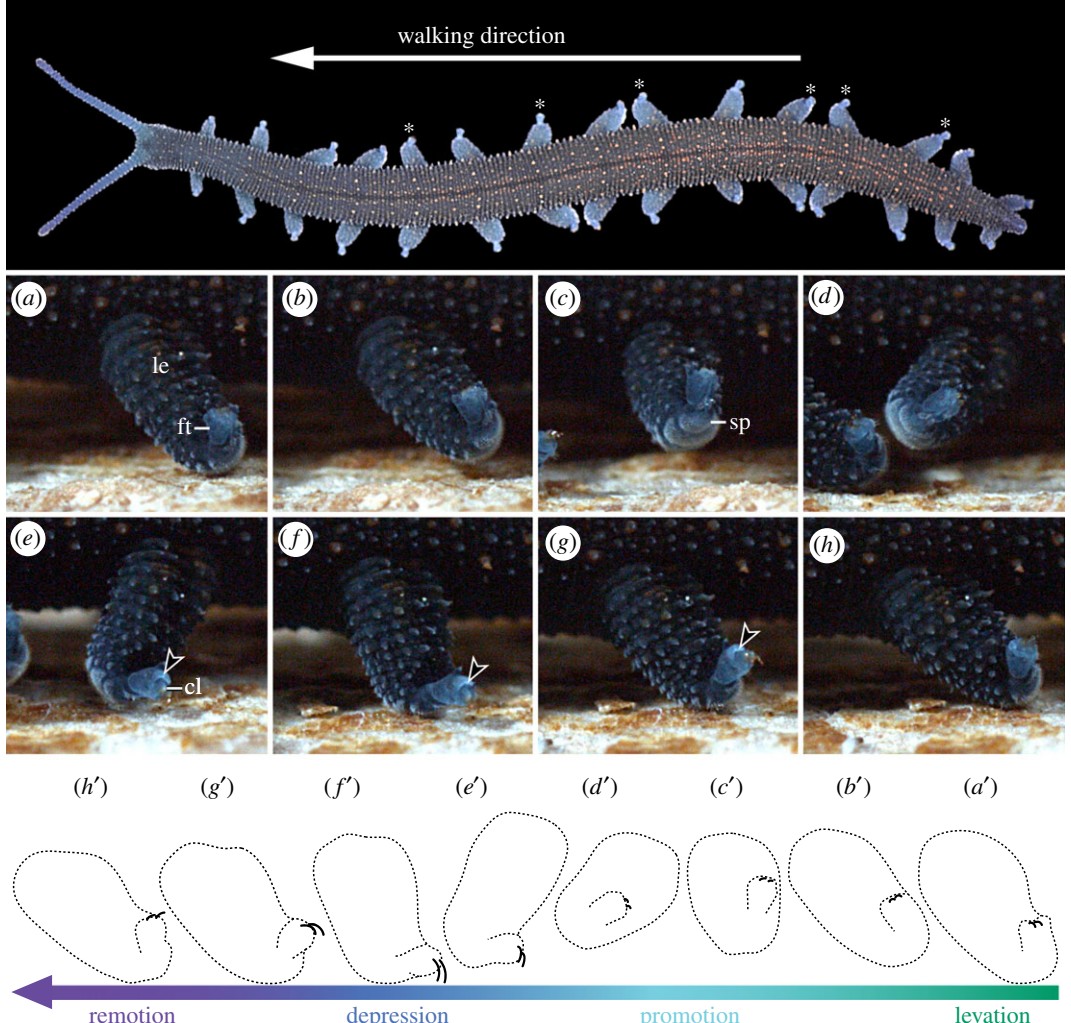

**Figure 6.** Lobopod movements during locomotion in *E. rowelli*. Photograph (top) and selected frames extracted from high-speed camera footage (500 fps) of walking specimen (*a*–*h*) with corresponding diagrams (*a′*–*h′*) showing major movements of individual lobopod. Asterisks in the top image highlight segments with synchronously moving lobopods. Gradient colour bar indicates continuous transition between major steps in locomotion. Arrowheads (in *e*–*g*) point to everted dorsal sac. Note complete retraction of claws at the end of levation (*b*) and fully protracted claws in the middle of depression (*f*).

located at their basis (arrowhead in figure 6*e*–*g*). Finally, the leg is remoted, passing back from an anterior to a posterior inclination of approximately 45°, while the body is pushed forward (figure 6*g,h,g′,h′*). At the beginning of leg remotion, both the spinous pads and the claws remain in contact with the substrate, but the claws are retracted and the eversible dorsal sac is inverted as the leg moves posteriorly, while the foot is levated (figure 6*g,h,g′,h′*).

## 4. Discussion

The myoanatomy of onychophoran lobopods has been described inconsistently in the literature, in particular regarding the number, arrangement and function of the individual leg muscles (table 1) [24,27,28,31,33,34]. Three main reasons might account for these inconsistencies: (i) interspecific variation, as different species were studied by different authors; (ii) deviating nomenclature and interpretation of muscles and their function; and (iii) shortcomings of the mainly histological methods used and the lack of high-speed video recordings and 3D reconstruction tools at that time. While one cannot rule out *per se* the interspecific variation, this does not seem to be the only reason for the discrepancies in previous reports, as at least two studies [24,31] focused on the same species (*Peripatus dominicae*) and yet resulted in substantially different descriptions of lobopodial muscles (table 1).

Deviating nomenclature is evident in several cases (table 1). For example, the leg depressor (muscle #2 of the present study) has been identified either as 'anterior depressor' [24] or 'ventral promotor' [27,28,31]. In order to clarify some of these inconsistencies and overcome previous methodological constraints, we analysed the muscular system of a mid-trunk lobopod from the onychophoran *E. rowelli* using high-resolution nano- [38] and micro-computed tomography [40].

Three-dimensional reconstruction revealed unmatched details of the position, extent and attachment sites of individual muscles and fibres. We identified 15 muscles associated with the leg of *E. rowelli* (electronic supplementary material, Movies S1–S4). This number clearly differs from those reported from other onychophoran species, including 6 in *Paraperipatus amboinensis* [33], 10 in *Peripatoides novaezealandiae* [28], 12 in *Peripatopsis* spp. [27] and 9 [24] or 22 [31] in *Peripatus dominicae*. A closer look at the descriptions revealed that 11 out of 15 muscles identified herein had already been documented previously, with the most overlap between the peripatopsid *E. rowelli* and the peripatid *P. dominicae* [31] (table 1). This suggests that at least these 11 muscles might be conserved among onychophorans and were present in the last common ancestor of Peripatidae and Peripatopsidae. However, we cannot exclude that the remaining four muscles identified in *E. rowelli*, including the anterior (#5) and posterior (#6) leg rotators, the leg flexor (#9) and the bridge constrictor (#14), might have been overlooked in other species studied, as they are fairly small, consist of a relatively low number of fibres and are located in the distalmost portion of the leg where the musculature is particularly dense and intricate. It is worth noting that 13 formerly described muscles, mostly reported by Birket-Smith [31], could not be identified or unambiguously assigned to any of the muscles in our datasets (table 1). Hence, their existence is uncertain. A detailed study of myoanatomy in a representative of Peripatidae might help to clarify this issue and to reconstruct the complete set of lobopodial muscles in the last common ancestor of Onychophora.

Another controversy surrounds the functional myoanatomy of the onychophoran leg. Previous authors used a divergent nomenclature for the individual leg muscles (table 1), which resulted in different scenarios to explain the operation of the onychophoran lobopod. For example, the muscle identified here as leg levator (#1) has mainly been interpreted as leg remotor in the literature (table 1) [24,27,28,31,35]. Nevertheless, the anatomy and anterodorsal position of this muscle within the leg speaks against its possible role to produce forward propulsion. The same applies to the leg depressor (#2), which was previously interpreted as a promotor muscle [27,28,31,35]. Interestingly, the levation and depression functions were formerly attributed to the somatic musculature [27,28,31,33,35], most likely because earlier authors interpreted the 'extrinsic' and 'intrinsic' portions of a single leg muscle as separate functional units. Apart from the present study, the only report of leg levator and depressor muscles in Onychophora [24] refers to structures that could not be identified in our dataset; they may thus not correspond to the muscles of the same name characterized herein (table 1).

To improve the nomenclature and to better understand the operational principles of the onychophoran lobopod, we combined 3D reconstructions of leg myoanatomy with the examination of high-speed camera recordings of individual legs in walking specimens of *E. rowelli*. We found clear correspondences between our morphological data and the information extracted from the video footage. By considering the structure, position and attachment sites of the individual leg muscles, we were able to infer the potential locomotory role for 9 out of 15 identified muscles. These nine muscles include the leg levator (#1), the leg depressor (#2), the leg promotor (#3), the leg remotor (#4), the anterior leg rotator (#5), the posterior leg rotator (#6), the leg flexor (#9), the claw retractor (#12) and the foot depressor (#13). Some of these muscles may play additional roles not implied in their names. For example, the claw retractor (#12) may also levate the foot, whereas the leg promotor (#3) and the leg depressor (#2) might additionally be responsible for rotating the leg up to 180° in the anterior and posterior directions, respectively. The twisted shape of the latter two muscles supports their additional function as leg rotators. It is important to note that the anterior (#5) and the posterior (#6) leg rotators seem to be responsible only for a fine rotation of the distal leg portion, most likely adjusting the placement of the spinous pads on the irregular substrate during locomotion.

Apart from the nine muscles that might play a major role in the locomotory movements of the lobopod in *E. rowelli*, the remaining six muscles do not seem to be directly involved in locomotion but rather in stabilizing the leg and regulating the hydrostatic pressure within it. These muscles include the anteroproximal (#7) and the posteroproximal (#8) leg muscles, the anteroposterior (#10) and the dorsoventral (#11) septal muscles, the bridge constrictor (#14) and the foot constrictor (#15) (electronic supplementary material, Movies S2–S4). The structure and position of the anteroproximal (#7) and the posteroproximal (#8) leg muscles suggest that these muscles might be responsible for regulating the haemolymph flow into and out of the leg by changing the dimensions of the haemolymph channel leading into the trunk cavity. Previously, these two muscles were interpreted either as retractors of the

nephropore [31] or, in another study [27] in which only the anteroproximal leg muscle (#7) was identified, it was assumed to prevent the leg from being excessively blown outwards when the hydrostatic pressure increases inside. On the other hand, the haemolymph flow into and out of the leg was believed to be controlled by the oblique trunk musculature [27]. However, our 3D reconstructions do not support any of these functions of the two muscles, as we found no evidence for their association with the nephropore (electronic supplementary material, figure S6a,b) or the existence of additional, oblique trunk musculature associated with the lobopod. The anteroproximal leg muscle (#7) also does not seem to play a role in the stabilization of the leg, as revealed by its position and reduced number of fibres.

The anteroposterior (#10) and the dorsoventral (#11) septal muscles have been identified previously [24,27,28,31], but their function remained unclear. Based on their anatomical characteristics, we suggest that these muscles might be responsible for stabilizing the lobopod along its two main axes as well as holding non-muscular structures in place (electronic supplementary material, Movie S3). In addition, they might be involved in hydrostatic control, assuming that their contraction would affect the haemolymph pressure inside the leg. Since the fibres of the septal muscles are more or less loose and do not form a dense sheet, we assume that the haemolymph may still flow freely through these muscles and is exchanged between the four blood compartments (I–IV). Consequently, the four blood compartments may exhibit the same hydrostatic pressure during the operation of the leg.

Finally, the bridge constrictor (#14) may function as a valve controlling the haemolymph flow through the bridge linking the foot with the leg. This muscle, together with the ring-shaped foot constrictor (#15), may be responsible for the protraction of the claws—the only movement of the onychophoran leg solely induced by the hydrostatic pressure [27,37]. Once the bridge constrictor (#14) closes the leg-to-foot connection, circular fibres of the foot constrictor (#15) might contract, thus increasing the hydrostatic pressure inside the foot and pumping the fluid into the dorsal eversible sac (electronic supplementary material, Movie S3). This may result in the eversion of the dorsal sac and the protraction of the two claws [37]. In contrast with a previous report [33], we did not find any evidence for the existence of antagonistic adductor and remotor muscles associated with the claws. Rather, we suggest that both claws are retracted by the claw retractor (#12), thus causing a passive inversion of the dorsal sac.

In the light of the new findings, it would be important to generate comparative myoanatomical data for the onychophoran jaws and slime papillae—highly modified limb derivatives belonging to the second and third head segments, respectively [9,47]. A recent study [37], in which serial homology of sclerotized parts of the claws and the jaws has been demonstrated, described eight muscles associated with each jaw. In this case, one would expect that at least some of the jaw and claw/leg muscles are also homologous, but this hypothesis still has to be tested. The same applies to the slime papillae, given that important morphological landmarks, such as sclerotized parts, are largely missing in the slime papilla segment and may hamper the recognition of serially homologous elements [37,48]. It is also important to highlight that no study has ever attempted to depict the onychophoran slime papillae in detail and, to date, fundamental anatomical data are missing for this structure. Clarifying these aspects could shed light on the functional anatomy and evolution of the onychophoran jaws and slime papillae, which have arisen in the onychophoran lineage.

# 5. Conclusion

Onychophoran-like lobopodians (e.g. [1,11,16,30]) were arguably some of the first animals to use metameric limbs for locomotion. Hence, study of the onychophoran lobopods might provide insights into the evolution of locomotion in early animals. We have shown here that the myoanatomy of the onychophoran lobopod is strikingly complex in terms of the number and arrangement of muscle fibres. The high number of myofibrils revealed by 3D reconstruction might reflect the high number of motor neurons and neurites supplying each leg [49], suggesting an elaborate neural control of the lobopodial muscles. Given the striking morphological similarity of onychophorans to the fossil lobopodians, it is tempting to assume that a similar muscular system might have existed in the last common panarthropod ancestor. However, this hypothesis is difficult to test for two reasons. First, the fossil lobopodians show a great morphological diversity including a variety of species whose phylogenetic relationships to each other as well as to the extant panarthropods are still uncertain. While some species might be closely allied with onychophorans, others seem to be more closely related to tardigrades, arthropods or even panarthropods as a whole [3,30,50]. Second, only limited information is available on the muscular system of fossil lobopodians. The onychophoran-like lobopodian *Tritonychus phanerosarkus* indeed possessed muscles in the trunk [3] that resembled the fan-shaped lobopodial muscles of extant onychophorans. The fan-shaped

arrangement of muscle fibres, however, could have been due to the soft body surface of these animals, which otherwise would have been deformed during muscle contraction. 'Extrinsic' muscles reported from *Antennacanthopodia gracilis* [30] also suggest that an onychophoran-like organization was present in at least some lobopodians, but given the insufficient preservation of these fossils, virtually nothing can be deduced about the number and arrangement of individual muscles within their lobopods.

Detailed comparison of our findings with those from the other two extant panarthropod groups, the arthropods and tardigrades, is also problematic. The marine and limnoterrestrial tardigrades are the only other extant panarthropod group that shows unjointed limbs [21,22]. However, miniaturization in these animals seems to have led to a reduction in the individual muscles to single myocytes [21,51]. Although this renders a comparison with the onychophorans difficult, a recent study of myoanatomy revealed similar numbers of lobopodial muscles in the eutardigrade *Hypsibius exemplaris* equalling 14 in the first, 12 in the second, 11 in the third and 10 in the fourth lobopods [51]. Interestingly, these muscles are also arranged in the periphery of the leg like in onychophorans (electronic supplementary material, figure S8; [51])—a pattern that could have been inherited from their last common ancestor. Nevertheless, it is currently not possible to homologize the individual lobopodial muscles between the onychophorans and tardigrades, as considerable myoanatomical changes might have occurred along with miniaturization (in tardigrades [21]) and terrestrialization (in onychophorans [5]).

The same holds true for a comparison with arthropods, which in contrast with the fossil lobopodians and extant onychophorans and tardigrades have developed an exoskeleton (sclerotization) and jointed appendages (arthropodization). During the lobopodium-to-arthropodium transition, an elaborate lever-style system evolved in the arthropod lineage [5]. Along with the articulation of limbs, some of the ancestral muscles might have been multiplied or split and acquired new functions. This is supported by a typically higher number of leg muscles in arthropods. For example, 20 leg muscles have been reported from the horseshoe crab *Limulus polyphemus* [52] and 26 from the whip scorpion *Mastigoproctus giganteus* [53]. Given that neither the homology of the individual leg segments nor the corresponding muscles has been established among different arthropod subgroups [54,55], comparing the muscular systems between the arthropods and onychophorans would be even more difficult. A possible solution for clarifying the homology of individual leg muscles in onychophorans, tardigrades and arthropods would be to identify muscle-specific genes and compare their expression patterns across panarthropods. Recent studies of the NK cluster and NK-linked homeobox genes indeed revealed that at least seven genes (*NK1*, *NK4*, *MSX*, *LBX*, *TLX*, *Nedx* and *Hhex*) might be expressed in the anlagen of specific leg muscles in the onychophoran *E. rowelli* [56,57]. Clarifying the identity of these muscles in *E. rowelli* and analysing the expression of these genes in tardigrades and arthropods might shed light on their homology among these groups and the evolution of the muscular system in panarthropod legs.

Taken together, our study revealed unprecedented detail of the lobopodial musculature in the onychophoran *E. rowelli*, a member of the Peripatopsidae. Since there is discrepancy between our present and the previous findings (table 1), the question arises of whether or not this is due to a methodological artefact or represents real interspecific variability. To clarify this question, a member of the second major onychophoran subgroup, the Peripatidae, must first be studied using a similar approach. This would enable a reliable extrapolation across Onychophora as a whole. Likewise, acquiring similar sets of myoanatomical data from various arthropods may help to reconstruct the myoanatomy of lobopods in the last common ancestor of Panarthropoda. Only then it will be possible to infer the lobopodial organization in early panarthropods by avoiding an onychophoran-biased view.

Ethics. The experiments in this study did not require an approval by an ethical committee. All procedures in this investigation complied with international and institutional guidelines, including the guidelines for animal welfare as laid down by the German Research Foundation (DFG). Animals were collected under the permit nos. SL100159/ 2011 and SL101720/2016 issued by the NSW National Parks & Wildlife Service and exported under the permit nos. WT2012-8163 and PWS2016-AU-001023 obtained from the Department of Sustainability, Environment, Water, Population and Communities.

Data accessibility. Electronic supplementary material, movies showing the 3D reconstructed structures and functional morphology of the lobopod in *E. rowelli* are available within the Dryad Digital Repository: http://dx.doi.org/10. 5061/dryad.1gh5376 [58].

Authors' contributions. I.S.O., A.K., H.J. and G.M. designed the study; G.M. and F.P. contributed with materials; I.S.O., M.M., H.J. and A.K. acquired the data; I.S.O., A.K., H.J. and G.M. analysed and interpreted the data; I.S.O., A.K. and G.M. drafted the manuscript. All authors revised the drafts and gave final approval for publication.

Competing interests. The authors declare no competing interests.

Funding. Financial support came from the Conselho Nacional de Desenvolvimento Científico e Tecnológico (CNPq Brazil: grant no. 290029/2010-4) and the Zentrale Forschungsförderung of the University of Kassel (ZFF: grant no. 1970/2016) to I.S.O., from the German Research Foundation (DFG: grant no. Ma 4147/3-1, 7-1) to G.M. and from the Deutsches Elektronen-Synchrotron (DESY: grant no. I-20170069) to I.S.O., H.J. and G.M.

Acknowledgements. We are thankful to members of the Department of Zoology, University of Kassel, for their assistance with animal husbandry. Noel N. Tait is acknowledged for his help with the permits and Dave M. Rowell, Paul Sunnucks, Noel N. Tait, Franziska A. Franke, Sandra Treffkorn and Michael Gerth for their support with collecting the specimens. The authors thank Vladimir Gross (University of Kassel) for constructive comments and critical reading of the manuscript. We also thank Jörg U. Hammel (DESY) for his help with SRμCT. Two anonymous reviewers are cordially acknowledged for providing constructive comments on an early version of the manuscript. The staff of the Department of Sustainability, Environment, Water, Population and Communities of the Australian Government are gratefully acknowledged for providing collecting and export permits.

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
