## [Reviewer comments · Royal Society Open Science]

Review History

RSOS-191200.R0 (Original submission)

Review form: Reviewer 1

Is the manuscript scientifically sound in its present form?

Yes

Are the interpretations and conclusions justified by the results?

Yes

Is the language acceptable?

Yes

Do you have any ethical concerns with this paper?

No

Have you any concerns about statistical analyses in this paper?

No

Recommendation?

Accept with minor revision (please list in comments)

Comments to the Author(s)

This paper presents very high quality data on onychophoran leg morphology, derived from methods that are used for the first time to investigate onychophoran legs. The authors' results reveal incredibly detailed features of the internal and external anatomy of onychophoran legs. The careful analyses of leg muscles reveal their likely functions in locomotion. These results are integrated with an analysis of onychophoran locomotion, revealing how the different muscles interact during locomotion. The text includes very detailed descriptions of the morphology of the onychophoran leg, and the data in the figures provides an unprecedented view of the internal anatomy of a lobopodal leg. Additionally, the authors provide a comprehensive synthesis of the many interpretations of muscle anatomy in onychophoran legs, based on their new data. They also do an excellent job of discussing how their results relate to the broader picture of panarthropod appendage evolution. In sum, this is an excellent paper that will be of broad interest to biologists. I am very satisfied with the rigor of this study, and agree strongly with the conclusions. Therefore, I have no major comments to provide the authors. I do have a few minor comments that I hope that the authors will find useful.

Minor comments:

Throughout the results section, I recommend inserting abbreviations that are used in the figures the first time a structure is referred to. For example "The onychophoran trunk exhibits a pair of ventrolateral nerve cords (nc)...." This will be very helpful to readers, since you provide such an excellently detailed description of morphology.

Page 5, line 35 of proof: Haemocytes are referred to in Fig. 2e, but are not labeled in this panel. I recommend labeling them.

Page 5, line 38 of proof: Large collagen fibers are referred to in Figs. 2b, 3c, 3e, but are not labeled in these panels. I recommend labeling them.

Page 5, line ~48-49 of proof: Nephridial structures are referred to in Fig. 1d, 1f, but are not labeled in these panels. I recommend labeling them.

Review form: Reviewer 2**Is the manuscript scientifically sound in its present form?**

Yes

Are the interpretations and conclusions justified by the results?

Yes

Is the language acceptable?

Yes

Do you have any ethical concerns with this paper?

No

Have you any concerns about statistical analyses in this paper?

No

Recommendation?

Accept with minor revision (please list in comments)

Comments to the Author(s)

This is an exceptionally fine anatomical study. It draws on diverse imaging and analytical approaches to document the musculature of an onychophoran appendage, places the data in a functional morphological context, and contextualises the results in terms of previous attempts to do the same. It is elegant, meticulous and strikingly beautiful.

I really only have one (quite minor) issue to which I will devote this review. The abstract mentions that inferring the ancestral structure of a panarthropod lobopod is one of the reasons for conducting this study. This work stands on its own regardless of whether or not the authors get close to shedding light on the ancestral panarthropod.

To explain what I'm getting at, the Conclusion has a statement that I'd recommend modifying, "Given the striking morphological similarity of onychophorans to the fossil lobopodians, one can assume that a similar muscular system might have existed in the last common panarthropod ancestor". Fossil lobopodians are greatly varied, and some are more closely related to tardigrades or arthropods than to onychophorans, so it might be better to single out particular taxa that are likely most closely allied to onychophorans (such as *Antennacanthopodia*) for comparison rather than suggesting that all lobopodians and even the last common ancestor of Panarthropoda had much or all of the musculature of *Euperipatoides rowelli*. Adding to that, the variability in what has been described in lobopod myoanatomy in Onychophora may be, as the authors suggest, an artefact of incomplete or erroneous descriptions in previous studies. It could, however, also signal real variability, in which case any one species might not allow for a safe extrapolation across Onychophora as a whole (or more widely across Panarthropoda).

The authors are cautious about homologising any aspect of the musculature of onychophorans and tardigrades, allowing that similarities could be due to common ancestry or to convergence, biased by miniaturization in tardigrades. This is a bit problematic because it leads to being unable to make predictions about the last common ancestor of onychophorans and tardigrades and almost assumes that the ancestor should be read from one of them (Onychophora) alone. Likewise arthropods are noted to be highly modified, and again one is left feeling like it's being suggested that onychophorans are our best proxy for the last common ancestor of major panarthropod groups. Just as extant tardigrades and arthropods are modified from the last common ancestor of Panarthropoda, surely crown-group Onychophora have suites of characters related to terrestrial locomotion.

Decision letter (RSOS-191200.R0)

08-Aug-2019

Dear Dr de Sena Oliveira,

On behalf of the Editors, I am pleased to inform you that your Manuscript RSOS-191200 entitled "Functional morphology of a lobopod: Case study of an onychophoran leg" has been accepted for publication in Royal Society Open Science subject to minor revision in accordance with the referee suggestions. Please find the referees' comments at the end of this email.

The reviewers and handling editors have recommended publication, but also suggest some minor revisions to your manuscript. Therefore, I invite you to respond to the comments and revise your manuscript.

- Ethics statement

- Data accessibility

<http://datadryad.org/submit?journalID=RSOS&manu=RSOS-191200>

- Competing interests

- Authors' contributions

- Acknowledgements

- Funding statement

Because the schedule for publication is very tight, it is a condition of publication that you submit the revised version of your manuscript before 17-Aug-2019. Please note that the revision deadline will expire at 00.00am on this date. If you do not think you will be able to meet this date please let me know immediately.

Kind regards,

on behalf of Professor Brooke Flammang (Associate Editor) and Kevin Padian (Subject Editor)
openscience@royalsociety.org

Reviewer comments to Author:

Reviewer: 1
Comments to the Author(s)

This paper presents very high quality data on onychophoran leg morphology, derived from methods that are used for the first time to investigate onychophoran legs. The authors' results reveal incredibly detailed features of the internal and external anatomy of onychophoran legs. The careful analyses of leg muscles reveal their likely functions in locomotion. These results are integrated with an analysis of onychophoran locomotion, revealing how the different muscles interact during locomotion. The text includes very detailed descriptions of the morphology of the onychophoran leg, and the data in the figures provides an unprecedented view of the internal anatomy of a lobopodal leg. Additionally, the authors provide a comprehensive synthesis of the many interpretations of muscle anatomy in onychophoran legs, based on their new data. They also do an excellent job of discussing how their results relate to the broader picture of panarthropod appendage evolution. In sum, this is an excellent paper that will be of broad interest to biologists. I am very satisfied with the rigor of this study, and agree strongly with the conclusions. Therefore, I have no major comments to provide the authors. I do have a few minor comments that I hope that the authors will find useful.

Minor comments:

Throughout the results section, I recommend inserting abbreviations that are used in the figures the first time a structure is referred to. For example "The onychophoran trunk exhibits a pair of

ventrolateral nerve cords (nc)...” This will be very helpful to readers, since you provide such an excellently detailed description of morphology.

Page 5, line 35 of proof: Haemocytetes are referred to in Fig. 2e, but are not labeled in in this panel. I recommend labeling them.

Page 5, line 38 of proof: Large collagen fibers are referred to in Figs. 2b, 3c, 3e, but are not labeled in these panels. I recommend labeling them.

Page 5, line ~48–49 of proof: Nephridial structures are referred to in Fig. 1d, 1f, but are not labeled in these panels. I recommend labeling them.

Reviewer: 2

Comments to the Author(s)

This is an exceptionally fine anatomical study. It draws on diverse imaging and analytical approaches to document the musculature of an onychophoran appendage, places the data in a functional morphological context, and contextualises the results in terms of previous attempts to do the same. It is elegant, meticulous and strikingly beautiful.

I really only have one (quite minor) issue to which I will devote this review. The abstract mentions that inferring the ancestral structure of a panarthropod lobopod is one of the reasons for conducting this study. This work stands on its own regardless of whether or not the authors get close to shedding light on the ancestral panarthropod.

To explain what I’m getting at, the Conclusion has a statement that I’d recommend modifying, “Given the striking morphological similarity of onychophorans to the fossil lobopodians, one can assume that a similar muscular system might have existed in the last common panarthropod ancestor”. Fossil lobopodians are greatly varied, and some are more closely related to tardigrades or arthropods than to onychophorans, so it might be better to single out particular taxa that are likely most closely allied to onychophorans (such as *Antennacanthopodia*) for comparison rather than suggesting that all lobopodians and even the last common ancestor of Panarthropoda had much or all of the musculature of *Euperipatoides rowelli*. Adding to that, the variability in what has been described in lobopod myoanatomy in Onychophora may be, as the authors suggest, an artefact of incomplete or erroneous descriptions in previous studies. It could, however, also signal real variability, in which case any one species might not allow for a safe extrapolation across Onychophora as a whole (or more widely across Panarthropoda).

The authors are cautious about homologising any aspect of the musculature of onychophorans and tardigrades, allowing that similarities could be due to common ancestry or to convergence, biased by miniaturization in tardigrades. This is a bit problematic because it leads to being unable to make predictions about the last common ancestor of onychophorans and tardigrades and almost assumes that the ancestor should be read from one of them (Onychophora) alone. Likewise arthropods are noted to be highly modified, and again one is left feeling like it’s being suggested that onychophorans are our best proxy for the last common ancestor of major panarthropod groups. Just as extant tardigrades and arthropods are modified from the last common ancestor of Panarthropoda, surely crown-group Onychophora have suites of characters related to terrestrial locomotion.

Author's Response to Decision Letter for (RSOS-191200.R0)

See Appendix A.

Decision letter (RSOS-191200.R1)

09-Sep-2019

Dear Dr de Sena Oliveira,

I am pleased to inform you that your manuscript entitled "Functional morphology of a lobopod: Case study of an onychophoran leg" is now accepted for publication in Royal Society Open Science.

on behalf of Professor Brooke Flammang (Associate Editor) and Kevin Padian (Subject Editor)
openscience@royalsociety.org

Associate Editor Comments to Author (Professor Brooke Flammang):
Associate Editor: 1
Comments to the Author:
(There are no comments.)

Reviewer comments to Author:

Appendix A

U N I K A S S E L
V E R S I T Ä T

Fachbereich 10
Mathematik und
Naturwissenschaften

Dr. Ivo de Sena Oliveira
FG Zoologie

University of Kassel, FB 10 – Zoology; 34132 Kassel

Prof. Dr. Brooke Flammang and Prof. Dr. Kevin Padian
Editors of the *Royal Society Open Science*

e ivo.oliveira@uni-kassel.de
t 0049- 561 804 48 03
f 0049- 561 804 48 00

IBC – Raum 1303
Heinrich-Plett-Str. 40
34132 Kassel

19. August 2019

- **Revision of the manuscript entitled “Functional morphology of a lobopod: Case study of an onychophoran leg” by Ivo de Sena Oliveira and co-authors.**

Dear Prof. Dr. Brooke Flammang and Prof. Dr. Kevin Padian,

Thank you for accommodating a fast and efficient review of our manuscript. We were very pleased to read the positive responses of the referees and cordially thank them for their helpful and constructive comments. We have incorporated all their suggestions. Please find below our point-by-point responses (in blue), as well as our revised manuscript with modifications highlighted using the Track Changes tool.

We hope that our amendments fulfil your expectations and that you may now consider our manuscript suitable for publication in the *Royal Society Open Science*.

We are looking forward to hearing from you.

Sincerely,
Dr. Ivo de Sena Oliveira
(on behalf of co-authors)

Reviewer: 1

Comments to the Author(s)

This papers presents very high quality data on onychophoran leg morphology, derived from methods that are used for the first time to investigate onychophoran legs. The authors' results reveal incredibly detailed features of the internal and external anatomy of onychophoran legs. The careful analyses of leg muscles reveal their likely functions in locomotion. These results are integrated with an analysis of onychophoran locomotion, revealing how the different muscles interact during locomotion. The text includes very detailed descriptions of the morphology of the onychophoran leg, and the data in the figures provides an unprecedented view of the internal anatomy of a lobopodal leg. Additionally, the authors provide a comprehensive synthesis of the many interpretations of muscle anatomy in onychophoran legs, based on their new data. They also do an excellent job of discussing how their results relate to the broader picture of panarthropod appendage evolution. In sum, this is an excellent paper that will be of broad interest to biologists. I am very satisfied with the rigor of this study, and agree strongly with the conclusions. Therefore, I have no major comments to provide the authors. I do have a few minor comments that I hope that the authors will find useful.

We really appreciate the positive comments of the referee.

Minor comments:

Throughout the results section, I recommend inserting abbreviations that are used in the figures the first time a structure is referred to. For example "The onychophoran trunk exhibits a pair of ventrolateral nerve cords (nc)..." This will be very helpful to readers, since you provide such an excellently detailed description of morphology.

We agree with the reviewer and have added the abbreviations to the text as suggested.

Page 5, line 35 of proof: Haemocytes are referred to in Fig. 2e, but are not labeled in in this panel. I recommend labeling them.

Done (apparently Fig.1e is meant instead of Fig. 2e).

Page 5, line 38 of proof: Large collagen fibers are referred to in Figs. 2b, 3c, 3e, but are not labeled in these panels. I recommend labeling them.

Done.

Page 5, line ~48-49 of proof: Nephridial structures are referred to in Fig. 1d, 1f, but are not labeled in these panels. I recommend labeling them.

Done.

Reviewer: 2

Comments to the Author(s)

This is an exceptionally fine anatomical study. It draws on diverse imaging and analytical approaches to document the musculature of an onychophoran appendage, places the data in a functional morphological context, and contextualises the results in terms of previous attempts to do the same. It is elegant, meticulous and strikingly beautiful.

We are very glad of reading these positive comments.

I really only have one (quite minor) issue to which I will devote this review. The abstract mentions that inferring the ancestral structure of a panarthropod lobopod is one of the reasons for conducting this study. This work stands on its own regardless of whether or not the authors get close to shedding light on the ancestral panarthropod.

To explain what I'm getting at, the Conclusion has a statement that I'd recommend modifying, "Given the striking morphological similarity of onychophorans to the fossil lobopodians, one can assume that a similar muscular system might have existed in the last common panarthropod ancestor". Fossil lobopodians are greatly varied, and some are more closely related to tardigrades or arthropods than to onychophorans, so it might be better to single out particular taxa that are likely most closely allied to onychophorans (such as Antennacanthopodia) for comparison rather than suggesting that all lobopodians and even the last common ancestor of Panarthropoda had much or all of the musculature of *Euperipatoides rowelli*. Adding to that, the variability in what has been described in lobopod myoanatomy in Onychophora may be, as the authors suggest, an artefact of incomplete or erroneous descriptions in previous studies. It could, however, also signal real variability, in which case any one species might not allow for a safe extrapolation across Onychophora as a whole (or more widely across Panarthropoda).

The authors are cautious about homologising any aspect of the musculature of onychophorans and tardigrades, allowing that similarities could be due to common ancestry or to convergence, biased by miniaturization in tardigrades. This is a bit problematic because it leads to being unable to make predictions about the last common ancestor of onychophorans and tardigrades and almost assumes that the ancestor should be read from one of them (Onychophora) alone. Likewise arthropods are noted to be highly modified, and again one is left feeling like it's being suggested that onychophorans are our best proxy for the last common ancestor of major panarthropod groups. Just as extant tardigrades and arthropods are modified from the last common ancestor of Panarthropoda, surely crown-group Onychophora have suites of characters related to terrestrial locomotion.

We agree with all the points raised by the referee and have modified the Conclusion section of our manuscript accordingly.

Functional morphology of a lobopod: Case study of an onychophoran legIvo de Sena Oliveira^{1,2,†,*}, Andreas Kumerics^{1,*}, Henry Jahn¹, Mark Müller³, Franz Pfeiffer^{3,4} & Georg Mayer¹

Formatted: German

¹Department of Zoology, Institute of Biology, University of Kassel, Kassel, Germany²Departamento de Zoologia, Instituto de Ciências Biológicas, Universidade Federal de Minas Gerais, Belo Horizonte, Brazil³Chair of Biomedical Physics, Department of Physics & Munich School of Bioengineering, Technical University of Munich, Garching, Germany⁴Department of Diagnostic and Interventional Radiology, Klinikum rechts der Isar, Technical University of Munich, 81675, München, Germany**Keywords:** myoanatomy, musculature, velvet worm, Onychophora, tomography, 3D reconstruction.**1. Summary**

Segmental, paired locomotory appendages are a characteristic feature of Panarthropoda — a diversified clade of moulting animals that includes onychophorans (velvet worms), tardigrades (water bears), and arthropods. While arthropods acquired a sclerotised exoskeleton and articulated limbs, onychophorans and tardigrades possess a soft body and unjointed limbs called lobopods, which they inherited from Cambrian lobopodians. To date, the origin and ancestral structure of the lobopods and their transformation into the jointed appendages are all poorly understood. We therefore combined high-resolution computed tomography with high-speed camera recordings to characterise the functional anatomy of a trunk lobopod from the onychophoran *Euperipatoides rowelli*. Three-dimensional reconstruction of the complete set of muscles and muscle fibres as well as non-muscular structures revealed the spatial relationship and relative volumes of the muscular, excretory, circulatory, and nervous systems within the leg. Locomotory movements of individual lobopods of *E. rowelli* proved far more diverse than previously thought and might be governed by a complex interplay of fifteen muscles, including one promotor, one remotor, one levator, one retractor, two depressors, two rotators, one flexor and two constrictors as well as muscles for stabilisation and haemolymph control. We discuss the implications of our findings for understanding the evolution of locomotion in panarthropods.

2. Introduction

Panarthropoda comprises a diverse animal clade characterised, among other features, by segmental, paired locomotory appendages [1-4]. Extant panarthropods are classified into three major subclades including Onychophora (velvet worms), Tardigrada (water bears), and Arthropoda (spiders, centipedes, crustaceans, insects, and allies), the latter being the most species-rich animal taxon [5-8]. The evolutionary origin of panarthropods, however, is very old (~520 Ma) and deeply embedded in the Cambrian explosion [5], more specifically in a paraphyletic assemblage of extinct, marine animals commonly known as lobopodians (Greek *lobos* = lobe; *podos* = foot), in reference to their characteristic, unjointed limbs called lobopods (=lobopodia; sing. lobopodium) [6, 7, 9-11]. Interestingly, similar unjointed locomotory limbs are still found in present-day onychophorans and tardigrades, while this ancestral type of appendages has been transformed into articulated limbs (=arthropodia; sing. arthropodium) in the arthropod lineage along with the evolution of a sclerotised exoskeleton [5, 8, 12, 13]. The homology of the two major types of panarthropod limbs is supported by a similar proximodistal expression pattern of the leg gap genes *homothorax*, *extradenticle*, *dachshund* and *Distal-less* in onychophorans and arthropods [14].

To date, only little is known about the evolution of unjointed limbs in the panarthropod ancestor, and their transformation into the articulated limbs — a key innovation for the successful radiation and

*Shared first authorship.

†Author for correspondence (ivo.oliveira@uni-kassel).

diversification of arthropods [15] — is poorly understood. Studies of evolutionary functional morphology suggest that this transformation might have been accompanied by an increase in limb length observed during evolution of lobopodians [5, 16, 17]. According to this scenario, longer limbs required a more precise regulatory mechanism for movement, thus the role of extrinsic muscles in limb control may have expanded and sclerotised structures may have evolved on the limbs and body wall to stabilise the muscle attachment sites [5]. Once these attachment sites were in place, they enlarged progressively until they were finally able to shift function from stabilisation to protection, and to form articulations between them [5]. At the same time, a hydrostatic skeleton was gradually replaced by the lever-style muscular system observed in extant arthropods [5, 17]. Although plausible, this hypothesis is difficult to test [5], since the incomplete preservation of most fossil lobopodians found to date and in particular their soft tissues including the musculature [3, 17, 18] precludes verification of the proposed series of morphological transformations.

Deleted: accompanied

Inferring the early origin of walking limbs in panarthropods, however, is problematic. Since fossil data lack resolution for assessing important anatomical details of limbs in lobopodians (e.g. number, arrangement and attachment sites of leg muscles), it is neither possible to assert how legs in the last common ancestor of Panarthropoda may have looked like, nor to reconstruct the evolutionary events that may have led to the acquisition of this morphological trait [5]. Alternatively, data obtained from unjointed limbs of extant panarthropod taxa have been used to infer the evolution and putative functional morphology of the ancestral lobopodium [5, 19]. Among panarthropods with unjointed limbs, comparison with tardigrades is problematic because these animals underwent miniaturisation, due to which several morphological features, such as the heart and the circulatory system, have been either lost or strongly reduced (e.g., each tardigrade muscle consists of a single cell) [5, 20–22]. Onychophorans, on the other hand, are exclusively terrestrial but more similar in morphology and body size to the Cambrian lobopodians than any other extant panarthropod group [23], thus being commonly used as a model for understanding the evolution of limbs in Panarthropoda [3, 5, 17, 19, 24–28].

The onychophoran lobopods (also called oncopods) resemble those of Cambrian lobopodians in several respects, including their fleshy and unjointed nature, the overall shape, the presence of distal claws, and the existence of ventral pads functioning as walking soles [9, 23, 29, 30]. These similarities suggest that the functional morphology of walking limbs in Cambrian lobopodians may have been indeed comparable to that of extant onychophorans [5, 19]. However, the onychophoran leg itself has not yet been explored sufficiently and many aspects of its functional anatomy remain unclear. The few studies of the onychophoran lobopods mainly relied on traditional morphological techniques such as histological and semi-thin sections [24, 27, 28, 31–33], which are prone to reconstruction artefacts and hinder proper interpretation of complex systems such as lobopodial musculature. To our knowledge, three-dimensional (3D) reconstruction of the complete set of muscles and muscle fibres associated with the onychophoran leg has not been carried out, nor have the movements of individual onychophoran legs been well characterised, with only a few studies briefly describing the coordination of legs and different gait patterns [27, 34]. Consequently, the number, arrangement and putative function of individual muscles have been described inconsistently in the literature (summarised in Table 1), thus proposing contradictory scenarios for the operational principles of the onychophoran leg, especially concerning the spatial and functional relationships between the muscles and the role and regulation of the hydrostatic system within the leg [24, 27, 28, 31, 33].

To clarify the myoanatomy and to better understand the operational principles of a lobopod, we investigated the mid-trunk legs of the velvet worm *Euperipatoides rowelli* by combining high-resolution x-ray nano- and micro-computed tomography data with high-speed camera recordings. While nano- and micro-computed tomography allows resolutions down to 400 nm, providing enough detail to distinguish single bundles of muscle fibres [35–37], high-speed camera recordings enable an analysis of movements of individual legs. On the basis of the new data, we provide a 3D reconstruction of a complete onychophoran leg including its muscular, nervous, excretory and circulatory systems as well as cuticular structures such as the skin, claws and foot apodeme. These reconstructions reveal details of the spatial relationships and relative volumes of structures and cavities (=lacunar system) inside the leg. Analyses of high-speed camera footage shed light on the operation of the onychophoran leg during forward movement and show that the onychophoran locomotion is more complex and the locomotory movements more diverse than previously thought. In addition, we identified possible sources of previous inconsistencies and discuss the evolutionary implications of our findings in terms of evolution of locomotion in panarthropods.

3. Materials and Methods

3.1. Specimens

Specimens of the onychophoran species *Euperipatoides rowelli* Reid, 1996 (Peripatopsidae) were collected in the Tallaganda State Forest (New South Wales, Australia; 35°26'S, 149°33'E, 954 m) and kept in culture under controlled conditions as described elsewhere [38]. Selected specimens were analysed *in vivo* or anaesthetised

with chloroform vapour and prepared further for different analyses as described below. Animals were collected under the permit numbers SL100159/2011 and SL101720/2016 issued by the NSW National Parks & Wildlife Service and exported under the permit numbers WT2012-8163 and PWS2016-AU-001023 obtained from the Department of Sustainability, Environment, Water, Population and Communities.

3.2. Micro-computed tomography

The synchrotron radiation micro-computed tomography (SR μ CT) data were obtained from an adult female specimen of *E. rowelli* fixed overnight in paraformaldehyde (PFA; 4% in phosphate-buffered saline (= PBS), 0.1 M, pH 7.4), contrasted in osmium tetroxide (OsO₄, 2% in water overnight, Science Services GmbH, Munich, Germany), dehydrated in an ascending ethanol series, dried in a CPD 030 Critical Point Dryer (BAL-TEC AG, Balzers, Liechtenstein) and scanned at the beamline P05 of the storage ring PETRA III (Deutsches Elektronen-Synchrotron — DESY, Hamburg, Germany) as described by Jahn et al. [37].

3.3. Nano-computed tomography

The x-ray nano-computed tomography (nanoCT) dataset analysed herein was previously generated by Müller et al. [35] and consisted of 1200 z-stack images of a left leg from the mid-body of a female of *E. rowelli*. For nanoCT analyses, a newborn specimen was fixed and stored in 4 % formaldehyde in PBS for several weeks. After several washes in PBS, the specimen was cut into pieces each containing a single leg with a small part of the body wall. Thereafter, the pieces were contrasted in 1 % OsO₄ overnight, dehydrated in an ethanol series, critical point dried and mounted onto standardised nanoCT sample holders. Acquisition, processing and reconstruction of nanoCT data were carried out as described by Müller et al. [35].

3.4. Three-dimensional reconstruction and relative volume estimates

Different tissue types and structures were initially recognised in the dataset based on their specific grey values, texture and orientation within the leg. Segmentation of leg tissues and structures was carried out manually using the software Amira 6.0.1 (FEI Visualisation Sciences Group, Burlington, MA, USA). 3D volume renderings of segmented structures were generated using the software VGStudio Max 3.0 (Volume Graphics GmbH, Heidelberg, Germany). Estimates of the relative volumes were obtained by assessing the number of voxels in each segmented structure in relation to the total number of voxels of the entire leg. For that, regions of interest (ROI) were created in VGStudio to initially separate leg and body regions. Structures (or parts of them) lying within the leg were exported individually as image stacks, imported into the freeware FIJI v1.52j [39] and hollow regions were filled using the Process>Binary>Fill Holes option. The filled structures were imported into VGStudio to obtain their respective numbers of voxels.

3.5. Scanning electron microscopy

Adult female specimens of *E. rowelli*, as well as moulted skins preserved in 70 % ethanol were cut into small pieces containing four leg pairs. Body parts were transferred to distilled water through a descending ethanol series, fixed overnight in 4 % PFA and washed several times in PBS. Selected pieces were subsequently embedded in albumin-gelatin medium (Sigma-Aldrich, St. Louis, USA; 3.75 g in 10 ml distilled water and 0.5 g in 2.5 ml distilled water, respectively), cooled down at 4° C for four hours, postfixed in 10 % PFA overnight and sectioned using a vibratome (Micron HM 650V; Thermo Scientific, Walldorf, Germany). Body parts, skin pieces and sections were dehydrated in an ascending ethanol series, dried in a critical point dryer (CPD 030, BAL-TEC AG), mounted onto standardised stubs, coated with gold-palladium in a Polaron SC 7640 Sputter Coater (VG Microtech, East Grinstead, West Sussex, UK) and imaged in a Hitachi S4000 field emission scanning electron microscope (Hitachi High-Technologies, Europe GmbH, Düsseldorf, Germany) as described previously [40, 41].

3.6. High-speed camera analyses

The locomotion of living specimens of *E. rowelli* was recorded using a high-speed camera (Phantom Miro LC320S; Vision Research Phantom, Wayne, New Jersey, USA) equipped with a macro lens (105 mm) at 500 fps. Walking animals were recorded on a piece of tree bark moistened with distilled water to simulate the substrate of a typical onychophoran habitat. Motion sequences were analysed using the software Adobe Premiere Pro CS5.5 (Adobe Systems Incorporated, San Jose, USA).

3.7. Image processing and panel design

Final image and movie processing was performed with Adobe Photoshop CS5.1 and Adobe Premiere Pro CS5.5. Illustrations and panels were designed with Adobe Illustrator CS5.1 (Adobe Systems Incorporated, San Jose, USA).

Deleted: ... [1]

4. Results

4.1. Anatomy of the lobopod excluding the musculature

The leg (**le**) of *E. rowelli* exhibits an externally annulated cuticle with transverse rings of dermal papillae (**dp**) (figure 1a). The leg bears a distal foot equipped with a pair of sickle-shaped claws (**cl**) and three distal foot papillae (**fp**) (one anterior, one dorsal and one posterior). The foot is connected with the remaining portion of the leg via a triangular ventral bridge (**br**; figure 1a). The bridge shows a deep, longitudinal median furrow, which corresponds to the foot apodeme (**ap**; figure 1a–f). The paired claws appear either protracted (figure 2a) or retracted into the foot (figure 2b), depending on their condition during the fixation. In the fully retracted state, only the smooth tips of the claws remain outside the foot (figure 2c). When protracted, a conspicuous dorsal sac (**es**) becomes externally visible at the basis of the claws, while the three distal foot papillae become erected, with their sensory bristles pointing in the dorsal, anterior and posterior directions (arrows in figure 2a). When the claws are retracted, the eversible sac is inverted back into the foot and the distal foot papillae bend towards the claws, so that their sensory bristles point in the distal direction (arrow in figure 2b). On the ventrodorsal portion of the leg, three arch-shaped spinous pads (**sp**) occur next to the bridge (figure 1a). While the first and the second pads are adjacent to each other, the second and the third spinous pads are separated by a narrow, spineless integumentary fold (**if**; arrow in figure 1a; Supplementary figure 1a,b; Supplementary Movie 1). At the ventral leg basis, the nephridial opening (**no**; arrowhead in figure 1a,b,d,f) appears as an inconspicuous longitudinal slit between the surrounding dermal papillae.

The internal space of the leg is largely occupied by an elaborate lacunar system composed by channels and cavities, which are filled by haemolymph (=blood) and account for nearly half of the leg volume in a fixed specimen (Table 2; Supplementary Movie 2). These include: (i) numerous transverse channels beneath the transverse rings of dermal papillae (**tc**; figure 3a–c,e); (ii) the four major compartments I–IV separated by the anteroposterior and dorsoventral septal muscles (figure c–e); and (iii) the foot cavity (**fc**) surrounding the claw bases and extending into the dorsal eversible sac (figures 2c; 3a–c,e; Supplementary figure 2a). All these channels and cavities are confluent with each other and connected to the main body cavity via a narrow channel dorsolateral to the nerve cord (arrow in figure 3e). Among the four major compartments of the leg cavity (**lc**), the two dorsal ones (I and II) are the largest (figure 3c–e).

Two types of blood cells, including haemo- (**he**) and nephrocytes (**ne**), are found within the haemal spaces of the leg. The haemocytes appear as small, individual cells that occur randomly in the entire haemolymph system of the leg, including the foot (figures 1b,e,f; arrowheads in 2b,c; Supplementary figure 2b; Supplementary Movie 1). The nephrocytes, in contrast, are relatively larger and aggregated into clusters of three to six cells. The clusters of nephrocytes are mainly localised in the anterodorsal (Ist) blood compartment (Supplementary figure 2b), where they are not only attached to each other but also to the surrounding muscles and other tissues via a mesh of large collagen fibres (**cf**; figures 1b,e; 3e,f).

Deleted: c

Additional non-muscular structures of the lobopod include elements of the nervous and excretory systems (Supplementary figure 3). The onychophoran trunk (**tr**) exhibits a pair of ventrolateral nerve cords (**nc**) that are linked with each other by a series of ring (**rc**) and median (**mc**) commissures (figure 1b; see ref. [42] for details). The ring commissures are absent in the leg-bearing regions, which instead show prominent pairs of segmental nerves, the anterior and posterior leg nerves (**ln**); these nerves arise from the dorsolateral portion of the nerve cord and extend into each leg (figure 1b). The two leg nerves give rise to numerous fibres (not reconstructed herein) supplying the musculature and the epidermis. The excretory organ of the leg (=nephridium) is mainly located in the proximal part of the leg and consists of the proximal sacculus (**sa**) connected via a convoluted nephridial duct (**nd**) to the distal bladder (**nb**), which tapers into a short excretory duct associated with the nephridial opening (**arrowhead in figure 1b,d,f; figure 3c–f; Supplementary figure 3**). The nephridial duct extends from the sacculus into the trunk, then loops back into the leg cavity and finally joins the bladder (figure 1b,e).

Deleted: s

Deleted: -

Deleted: e

4.2. Myoanatomy of the lobopod

Segmentation and 3D reconstruction of individual bundles of muscle fibres revealed an intricate meshwork of 15 muscles within the lobopod of *E. rowelli* (Supplementary Movies 1–4). In the following, these muscles are numbered consecutively (#1–#15) and named according to their position within the leg and/or presumed function (Table 2; figure 4).

Leg levator (#1). This muscle forms a sheet, which embraces the posterodorsal (IInd) blood compartment (figures 1d; 3c,f; 4). Its proximal fibres project into the trunk and attach to the dorsolateral body wall, but the cuticle covering this region does not show any apodeme or apodeme-like structure (figure 5a,b; Supplementary figure 4). Relatively flat bundles of fibres belonging to the leg levator fan out in both dorsal (“extrinsic” portion) and ventral directions (“intrinsic” portion) and are associated mainly with the posterior leg region (figures 4; 5b). When followed dorsoventrally, the muscle undergoes a counterclockwise torsion of $\sim 90^\circ$ (figures 4; 5b). Distally, most fibres of the leg levator attach to the posteroventral surface of the leg and the spineless integumentary fold. However, a small subset of ventralmost fibres crosses the leg anteriorly and attaches to the anteroventral epidermis (figure 3f); another small subset of anteriormost fibres runs along the posterodorsal border of the leg and attaches distally to the epidermal folds between the transverse rings of the integument (figure 1d).

Leg depressor (#2). The leg depressor occupies the anteroventral portion of the leg (figures 3e,f; 4; 5a,b,d). The proximal fibres of this muscle run into the trunk towards the ventral midline (figure 5a,b,d), where a large set of fibres attaches to epidermal cells that form the ventral (vo) and pre-ventral (pv) organs—segmental, apodeme-like structures (figure 5a–d). Within the leg, the fibres of the leg depressor form a flat and dense muscle sheet that undergoes a $\sim 180^\circ$ torsion while crossing the anteroventral (IIIrd) blood compartment (figures 1d,f; 3f; 4; 5b,d). Some fibres extend in the posteroventral leg region and attach next to the distal margin of the nephridial opening and further distal between the first and the second spinous pad (figures 3e; 4; 5d). A small set of fibres fan out and attach to the anterodistal portion of the leg (figures 1e; 4).

Leg promotor (#3). This is the most prominent leg muscle (Table 2). It occupies most of the anterodorsal part of the leg and forms a sheet that covers the anterodorsal (Ist) blood compartment (figures 1c; 3c–f; 4; 5a,b,d). The “extrinsic” portion of the leg promotor is formed by fibres that attach along the dorsolateral body wall, not being associated with any type of apodeme-like structure (Supplementary figure 4). Near the leg basis, fibres belonging to the leg promotor become densely packed, forming a flat bundle that surrounds the anterodorsal blood compartment (figures 3c,e,f; 4; 5a,b). A small set of anteriormost extrinsic fibres surrounds the anterior half of the haemolymph channel opening into the body cavity and attaches to the ventral leg basis (figure 5a,b), while the remaining fibres fan out into the leg (figures 1c; 4). The numerous fibres composing the “intrinsic” portion of the leg promotor follow two distinct pathways: Anterior fibres project distally along the anterior leg region and attach ventrally to the spineless integumentary fold and the second spinous pad (figures 1c; 3f; 4; Supplementary figure 5a,b); posterior fibres run dorsomedially (figures 1c; 3c,e; 4) and attach to the dorsodistal leg region (figure 4).

Leg remotor (#4). This muscle occupies the ventroposterior portion of the leg. Proximally, its “extrinsic” fibres extend to the ventral midline of the trunk (figure 5b,d), where a small set of anteriormost fibres attach to the epidermal cells of the ventral organ (figure 5b–d) whereas their great majority fans out and attaches further posteriorly (figure 5b,d). When followed distally, the fibres of the leg remotor converge into a dense muscle that crosses the posteroventral (IVth) blood compartment as it undergoes a $\sim 90^\circ$ rotation (figures 1d, f; 3f; 4; 5b,d). The “intrinsic” portion of the leg remotor is composed of a small number of ventralmost fibres projecting distally and attaching to the posteroventral leg region (figure 1d) and numerous bundles of sparsely arranged fibres, which run medially towards the dorsal leg region and attach anterodorsally (figures 1d,e; 3f; 4).

Anterior leg rotator (#5). This is a short “intrinsic” muscle located distally in the anterior leg region (figures 1c; 4). Proximally, the parallel fibres of the anterior leg rotator embrace the anterodorsal (Ist) blood compartment, where they attach to the leg surface (figures 1c; 3c; 4). In its distal portion, fibres run ventrally and attach to the ridges of the spinous pads (figure 1c). This muscle is not associated with an apodeme-like structure.

Posterior leg rotator (#6). The posterior leg rotator is an “intrinsic” sheet-like muscle formed by a set of fibres surrounding the distal portion of the posterodorsal (IInd) blood compartment (figures 1c–e; 3c; 4). Ventrally, the fibres of the posterior leg rotator attach to the posterior edge of the spineless integumentary fold (arrow in figure 1a,d) as well as along the border of the second spinous pad. The proximal portion of this muscle attaches medially to the dorsal surface of the leg (figures 1d,e; 4). This muscle is not associated with an apodeme-like structure.

Anteroproximal (#7) and posteroproximal (#8) leg muscles. These two muscles are formed by small sets of densely arranged fibres that dorsoventrally cross the main channel connecting the cavities of the leg and the trunk (figures 4; 5a,b). The “extrinsic” portion of these muscles extends to the dorsolateral body wall of the trunk, while the “intrinsic” portion attaches to the ventral surface of the leg basis. More specifically, the ventral fibres of the anteroproximal muscle (#7) project anteriorly towards the leg depressor (#2; figure 5b), whereas those of the posteroproximal muscle (#8) arch posteriorly and attach further posteriorly (figure 5b).

Neither of these two muscles are associated with the nephridium or apodeme-like structures (Supplementary figure 6a,b).

Leg flexor (#9). The flexor is an “intrinsic” muscle embracing the posteroventral (IVth) blood compartment (figures 1d; 4). Distally, its fibres are densely packed and mainly attach to the ridge of the third spinous pad, with only a few of them connecting to the ridge of the spineless integumentary fold (figures 1d; 4). The fibres of the proximal part of the leg flexor fan out and attach to the posterior surface of the leg (figures 1d; 4). This muscle is not associated with an apodeme-like structure.

Anteroposterior septal muscle (#10). This muscle is formed by numerous parallel, “intrinsic” fibres that run along the anteroposterior axis of the leg and form a septum-like sheet, which subdivides the leg cavity into dorsal (Ist + IIInd) and ventral (IIIrd + IVth) blood compartments (figures 1e; 3e,f; 4). Bundles of fibres belonging to this muscle appear spaced anteriorly and posteriorly, i.e. near their attachment points to the anterior and posterior walls of the leg, whereas the fibres converge into a compact muscle medially (figure 4). The anteroposterior septal muscle extends nearly through the entire leg from the level of the nephropore to the level of the second spinous pad and borders ventrally the nephridial bladder and the two leg nerves (figures 1e; 3e,f; 4). It is not associated with an apodeme-like structure.

Dorsoventral septal muscle (#11). This muscle appears as a sheet of more or less parallel fibres that are more compact in the distal leg region and span the leg dorsoventrally, thus subdividing its cavity medially into anterior (Ist + IIIrd) and posterior (IIInd + IVth) blood compartments (figures 1c–f; 3c,f; 4). The dorsoventral septal muscle extends from the level of the nerve cord (proximal limit) to the second spinous pad (distal limit) (figures 1e,f; 3c; 4). A small subset of proximal fibres projects dorsally into the trunk, where they are associated with the lateral body wall (“extrinsic” portion), and ventrally into the leg, where they attach to the wall of the leg basis next to the nephridial opening (figure 1c–f). However, most fibres belonging to this muscle are still located within the leg (“intrinsic” portion), with their attachment points distributed along the dorsal and ventral proximodistal midlines of the leg (figures 1e,f; 3c; 4). This muscle is not associated with an apodeme-like structure.

Claw retractor (#12). The claw retractor is a dense, prominent muscle that runs along the ventral area of the anterodorsal (Ist) blood compartment and crosses the bridge into the foot (figures 1e; 3e; 4; Supplementary figure 5). Distally, the claw retractor connects to the dorsal rim of each claw (figures 2c; 3e; 4; Supplementary figure 7a), while proximally, this muscle fuses with the leg promotor (#3) and their fibres become indistinct from each other (Supplementary figure 7b). The claw retractor possibly attaches along the dorsolateral body wall together with the leg promotor (#3). Several bundles of fibres branch off along the retractor and project ventrally (Supplementary figure 7a,b). The two most prominent bundles include a distal branch, which attaches to the spineless integumentary fold, and a proximal “extrinsic” branch, which projects into the trunk and attaches to the ventrolateral body wall (figures 3e; 4). Additional sets of fibres branch off this muscle in a helical pattern and attach to the ventral wall of the leg, thus giving the claw retractor a twisted appearance (figure 4; Supplementary figure 7a,b). This muscle is not associated with an apodeme-like structure.

Foot depressor (#13). This “intrinsic” muscle is located ventrally in the distal portion of the leg (figures 1c–e; 2c; 3e; 4). It appears as a flat muscle and attaches distally to the foot apodeme located in the bridge (figures 1d; 2c; 3e). Its proximal fibres fan out and attach to the edges of the third spinous pad and the spineless integumentary fold (figure 1a,d).

Bridge constrictor (#14). The bridge constrictor is the least prominent leg muscle. It is composed of a few “intrinsic” fibres located dorsal to the bridge (figures 1f; 2c; 4). Its fibres are oriented anteroposteriorly and attach to the anterior and posterior walls of the leg. This muscle is not associated with an apodeme-like structure.

Foot constrictor (#15). This muscle consists of more or less parallel, circular fibres located underneath the surface of the foot (figure 4; Supplementary figure 2a). The ring fibres run perpendicularly to the main axis of the foot and surround nearly the entire foot cavity including that of the retracted eversible dorsal sac (figures 1c–f; 2c; 3c,e; 4). Some rings appear incomplete ventrally, while others show a denser ventral than dorsal arrangement (Supplementary figure 2a).

4.3. Relative volumes of structures and cavities within the lobopod

Relative volume estimates (Table 2) reveal that the haemolymph space occupies nearly half of the lobopod (45.71 %), followed by the epidermis and connective tissue including extracellular matrix and large collagen fibres (25.47 %), and muscular (14.78 %) and non-muscular structures (14.04 %). Among non-muscular structures, elements of the excretory system (5.96 %) and the cuticle (5.79 %) are the largest. The most voluminous muscle (taking only the “intrinsic” muscular portions into account) is the leg promotor (#3; occupying 4.16 %), while the least voluminous muscle is the bridge constrictor (#14; occupying only 0.03 % of the total leg volume).

4.4. Locomotory movements of individual lobopods during forward walking

Specimens of *E. rowelli* are able to walk either forward or backward. They exhibit different types of gaits, depending on structure and properties of the substrate, experimental conditions, and whether or not the specimen is stressed. During locomotion, even different body regions of the same specimen may show different gait patterns, in which the two lobopods of the same segment are moved either synchronously or out of phase (figure 6). High-speed camera recordings of walking specimens in lateral view demonstrate that the movements of individual lobopods during forward walking involve seven major operational modes: (1) levation or depression; (2) promotion or remotion; (3) stretching, contracting and/or arching; (4) anterior and posterior rotation (up to $\sim 180^\circ$); (5) fine rotation of the distal leg portion (bearing the spinous pads); (6) foot levation or depression; and (7) claw protraction or retraction (Supplementary Movie 5).

Considering an individual lobopod of *E. rowelli* walking forward on a piece of wood (simulating the substrate of a typical microhabitat), its movements can be characterised as follows (figure 6a–h, a'–h'; Supplementary Movie 5). At the starting position (figure 6a, a'), the leg is inclined at $\sim 45^\circ$ posteriorly, the spinous pads are in contact with the substrate, the foot is levated and the claws are retracted. During leg levation, the leg is lifted from the substrate maintaining its initial 45° inclination relative to the body (figure 6b, b'). Thereafter, the leg is promoted above the substrate, rotates at $\sim 90^\circ$, and passes from a posterior to an anterior inclination of $\sim 45^\circ$ (figure 6b–d, b'–d'). During the levation and promotion of the leg, the foot remains levated. The leg depression begins when the spinous pads are facing anteriorly (figure 6d, d') and the leg is then pushed towards the ground until the posterior margins of the spinous pads touch the substrate (figure 6e, e', e'). At this point, the foot is depressed and the claws are protracted, hooking onto the substrate. The protraction of the claws is accompanied by the eversion of the dorsal sac located at their basis (arrowhead in figure 6e–g). Finally, the leg is remoted, passing back from an anterior to a posterior inclination of $\sim 45^\circ$, while the body is pushed forward (figure 6g, h, g', h'). At the beginning of leg remotion, both the spinous pads and the claws remain in contact with the substrate, but the claws are retracted and the eversible dorsal sac is inverted as the leg moves posteriorly, while the foot is levated (figure 6g, h, g', h').

5. Discussion

The myoanatomy of onychophoran lobopods has been described inconsistently in the literature, in particular regarding the number, arrangement and function of the individual leg muscles (Table 1) [24, 27, 28, 31, 33, 34]. Three main reasons might account for these inconsistencies: (1) interspecific variation, as different species were studied by different authors; (2) deviating nomenclature and interpretation of muscles and their function; and (3) shortcomings of the mainly histological methods used and the lack of high-speed video recordings and 3D reconstruction tools at that time. While one cannot rule out *per se* the interspecific variation, this does not seem to be the only reason for the discrepancies in previous reports, as at least two studies [24, 31] focused on the same species (*Peripatus dominicae*) and yet resulted in substantially different descriptions of lobopodial muscles (Table 1). Deviating nomenclature is evident in several cases (Table 1). For example, the leg depressor (muscle #2 of the present study) has been identified either as “anterior depressor” [24] or “ventral promotor” [27, 28, 31]. In order to clarify some of these inconsistencies and overcome previous methodological constraints, we analysed the muscular system of a mid-trunk lobopod from the onychophoran *E. rowelli* using high-resolution nano- [35] and micro-computed tomography [36].

3D reconstruction revealed unmatched details of the position, extent, and attachment sites of individual muscles and fibres. We identified fifteen muscles associated with the leg of *E. rowelli* (Supplementary Movies 1–4). This number clearly differs from those reported from other onychophoran species, including six in *Paraperipatus amboinensis* [33], ten in *Peripatoides novaezealandiae* [28], twelve in *Peripatopsis* spp. [27], and nine [24] or twenty-two [31] in *Peripatus dominicae*. A closer look at the descriptions revealed that eleven out of fifteen muscles identified herein had already been documented previously, with the most overlap between the peripatopsid *E. rowelli* and the peripatid *P. dominicae* [31] (Table 1). This suggests that at least these eleven muscles might be conserved among onychophorans and were present in the last common ancestor of Peripatidae and Peripatopsidae. However, we cannot exclude that the remaining four muscles identified in *E. rowelli*, including the anterior (#5) and posterior (#6) leg rotators, the leg flexor (#9) and the bridge constrictor (#14), might have been overlooked in other species studied, as they are fairly small, consist of a relatively low number of fibres and are located in the distal-most portion of the leg where the musculature is particularly dense and intricate. It is worth noting that thirteen formerly described muscles, mostly reported by Birket-Smith [31], could not be identified or unambiguously assigned to any of the muscles in our datasets (Table 1). Hence, their existence is uncertain. A detailed study of myoanatomy in a

representative of Peripatidae might help to clarify this issue and to reconstruct the complete set of lobopodial muscles in the last common ancestor of Onychophora.

Another controversy surrounds the functional myoanatomy of the onychophoran leg. Previous authors used a divergent nomenclature for the individual leg muscles (Table 1), which resulted in different scenarios to explain the operation of the onychophoran lobopod. For example, the muscle identified here as leg levator (#1) has mainly been interpreted as leg remotor in the literature (Table 1) [24, 27, 28, 31, 43]. Nevertheless, the anatomy and anterodorsal position of this muscle within the leg speaks against its possible role to produce forward propulsion. The same applies to the leg depressor (#2), which was previously interpreted as a promotor muscle [27, 28, 31, 43]. Interestingly, the levation and depression functions were formerly attributed to the somatic musculature [27, 28, 31, 33, 43], most likely because earlier authors interpreted the “extrinsic” and “intrinsic” portions of a single leg muscle as separate functional units. Apart from the present study, the only report of leg levator and depressor muscles in Onychophora [24] refers to structures that could not be identified in our dataset; they may thus not correspond to the muscles of the same name characterised herein (Table 1).

To improve the nomenclature and to better understand the operational principles of the onychophoran lobopod, we combined 3D reconstructions of leg myoanatomy with the examination of high-speed camera recordings of individual legs in walking specimens of *E. rowelli*. We found clear correspondences between our morphological data and the information extracted from the video footage. By considering the structure, position and attachment sites of the individual leg muscles, we were able to infer the potential locomotory role for nine out of fifteen identified muscles. These nine muscles include the leg levator (#1), the leg depressor (#2), the leg promotor (#3), the leg remotor (#4), the anterior leg rotator (#5), the posterior leg rotator (#6), the leg flexor (#9), the claw retractor (#12), and the foot depressor (#13). Some of these muscles may play additional roles not implied in their names. For example, the claw retractor (#12) may also levate the foot, whereas the leg promotor (#3) and the leg depressor (#2) might additionally be responsible for rotating the leg up to 180° in the anterior and posterior directions, respectively. The twisted shape of the latter two muscles supports their additional function as leg rotators. It is important to note that the anterior (#5) and the posterior (#6) leg rotators seem to be responsible only for a fine rotation of the distal leg portion, most likely adjusting the placement of the spinous pads on the irregular substrate during locomotion.

Apart from the nine muscles that might play a major role in the locomotory movements of the lobopod in *E. rowelli*, the remaining six muscles do not seem to be directly involved in locomotion but rather in stabilising the leg and regulating the hydrostatic pressure within it. These muscles include the anteroproximal (#7) and the posteroproximal (#8) leg muscles, the anteroposterior (#10) and the dorsoventral (#11) septal muscles, the bridge constrictor (#14), and the foot constrictor (#15) (Supplementary Movies 2–4). The structure and position of the anteroproximal (#7) and the posteroproximal (#8) leg muscles suggest that these muscles might be responsible for regulating the haemolymph flow into and out of the leg by changing the dimensions of the haemolymph channel leading into the trunk cavity. Previously, these two muscles were interpreted either as retractors of the nephropore [31] or, in another study [27] in which only the anteroproximal leg muscle (#7) was identified, it was assumed to prevent the leg from being excessively blown outwards when the hydrostatic pressure increases inside. On the other hand, the haemolymph flow into and out of the leg was believed to be controlled by the oblique trunk musculature [27]. However, our 3D reconstructions do not support any of these functions of the two muscles, as we found no evidence for their association with the nephropore (Supplementary figure 6a,b) or the existence of additional, oblique trunk musculature associated with the lobopod. The anteroproximal leg muscle (#7) also does not seem to play a role in the stabilisation of the leg, as revealed by its position and reduced number of fibres.

The anteroposterior (#10) and the dorsoventral (#11) septal muscles have been identified previously [24, 27, 28, 31], but their function remained unclear. Based on their anatomical characteristics, we suggest that these muscles might be responsible for stabilising the lobopod along its two main axes as well as holding non-muscular structures in place (Supplementary Movie 3). In addition, they might be involved in hydrostatic control, assuming that their contraction would affect the haemolymph pressure inside the leg. Since the fibres of the septal muscles are more or less loose and do not form a dense sheet, we assume that the haemolymph may still flow freely through these muscles and is exchanged between the four blood compartments (I–IV). Consequently, the four blood compartments may exhibit the same hydrostatic pressure during the operation of the leg.

Finally, the bridge constrictor (#14) may function as a valve controlling the haemolymph flow through the bridge linking the foot with the leg. This muscle, together with the ring-shaped foot constrictor (#15), may be responsible for the protraction of the claws – the only movement of the onychophoran leg solely induced by the hydrostatic pressure [27, 44]. Once the bridge constrictor (#14) closes the leg-to-foot connection, circular fibres of the foot constrictor (#15) might contract, thus increasing the hydrostatic pressure inside the foot and pumping the fluid into the dorsal eversible sac (Supplementary Movie 3). This may result in the eversion of

the dorsal sac and the protraction of the two claws [44]. In contrast to a previous report [33], we did not find any evidence for the existence of antagonistic adductor and remotor muscles associated with the claws. Rather, we suggest that both claws are retracted by the claw retractor (#12), thus causing a passive inversion of the dorsal sac.

In the light of the new findings, it would be important to generate comparative myoanatomical data for the onychophoran jaws and slime papillae — highly modified limb derivatives belonging to the second and third head segments, respectively [9, 45]. A recent study [44], in which serial homology of sclerotised parts of the claws and the jaws has been demonstrated, described eight muscles associated with each jaw. In this case, one would expect that at least some of the jaw and claw/leg muscles are also homologous, but this hypothesis still has to be tested. The same applies to the slime papillae, given that important morphological landmarks, such as sclerotised parts, are largely missing in the slime papilla segment and may hamper the recognition of serially homologous elements [44, 46]. It is also important to highlight that no study has ever attempted to depict the onychophoran slime papillae in detail and, to date, fundamental anatomical data are missing for this structure. Clarifying these aspects could shed light on the functional anatomy and evolution of the onychophoran jaws and slime papillae, which have arisen in the onychophoran lineage.

6. Conclusion

Onychophoran-like lobopodians (e.g. [1, 11, 16, 30]) were arguably one of the first animals to use metameric limbs for locomotion. Hence, study of the onychophoran lobopods might provide insights into the evolution of locomotion in early animals. We have shown here that the myoanatomy of the onychophoran lobopod is strikingly complex in terms of the number and arrangement of muscle fibres. The high number of myofibrils revealed by 3D reconstruction might reflect the high number of motor neurons and neurites supplying each leg [47], suggesting an elaborate neural control of the lobopodial muscles. Given the striking morphological similarity of onychophorans to the fossil lobopodians, it is tempting to assume that a similar muscular system might have existed in the last common panarthropod ancestor. However, this hypothesis is difficult to test for two reasons. First, the fossil lobopodians show a great morphological diversity including a variety of species whose phylogenetic relationships to each other as well as to the extant panarthropods are still uncertain. While some species might be closely allied with onychophorans, others seem to be more closely related to tardigrades, arthropods, or even panarthropods as a whole [3, 30, 48]. Second, only limited information is available on the muscular system of fossil lobopodians. The onychophoran-like lobopodian †*Tritomychus phanerosarkus* indeed possessed muscles in the trunk [3] that resembled the fan-shaped lobopodial muscles of extant onychophorans. The fan-shaped arrangement of muscle fibres, however, could have been due to the soft body surface of these animals, which otherwise would have been deformed during muscle contraction. “Extrinsic” muscles reported from †*Antennacanthopodia gracilis* [30] also suggest that an onychophoran-like organisation was present in at least some lobopodians, but given the insufficient preservation of these fossils, virtually nothing can be deduced about the number and arrangement of individual muscles within their lobopods.

Detailed comparison of our findings with those from the other two extant panarthropod groups, the arthropods and tardigrades, is also problematic. The marine and limnoterrestrial tardigrades are the only other extant panarthropod group that shows unjointed limbs [21, 22]. However, miniaturisation in these animals seems to have led to a reduction of the individual muscles to single myocytes [21, 49]. Although this renders a comparison with the onychophorans difficult, a recent study of myoanatomy revealed similar numbers of lobopodial muscles in the eutardigrade *Hypsibius exemplaris* equalling fourteen in the first, twelve in the second, eleven in the third, and ten in the fourth lobopods [49]. Interestingly, these muscles are also arranged in the periphery of the leg like in onychophorans (Supplementary figure 8; [49]) — a pattern that could have been inherited from their last common ancestor. Nevertheless, it is currently not possible to homologue the individual lobopodial muscles between the onychophorans and tardigrades, as considerable myoanatomical changes might have occurred along with miniaturisation (in tardigrades [21]) and terrestrialisation (in onychophorans [5]).

The same holds true for a comparison with arthropods, which in contrast to the fossil lobopodians and extant onychophorans and tardigrades have developed an exoskeleton (sclerotisation) and jointed appendages (arthropodisation). During the lobopodium-to-arthropodium transition, an elaborate lever-style system evolved in the arthropod lineage [5]. Along with the articulation of limbs, some of the ancestral muscles might have been multiplied or split and acquired new functions. This is supported by a typically higher number of leg muscles in arthropods. For example, twenty leg muscles have been reported from the horseshoe crab *Limulus polyphemus* [50] and twenty-six from the whip scorpion *Mastigoproctus giganteus* [51]. Given that neither the homology of the individual leg segments nor the corresponding muscles has been

Deleted: one can

Deleted: ;

Deleted: Although only limited information is available on the myoanatomy of fossil lobopodians, these animals might have possessed a well-developed leg musculature with “extrinsic” portions [3, 30], which might correspond to the four most prominent leg muscles (#1–#4) of *E. ravelli*. Even the fan-shaped nature of the individual muscles might have existed in the extinct, marine lobopodians [3, 18], like in extant, terrestrial onychophorans, probably due to the soft body surface of these animals, which otherwise would have been deformed during muscle contraction. ... [2]

Deleted: are the marine and limnoterrestrial tardigrades

Deleted: of tardigrades might have been reduced

Deleted: muscle cells due to a miniaturisation of these animals

Deleted: or, alternatively, arose convergently due to morphological constraints imposed by their soft cuticle

Deleted: seems

Deleted: im

Deleted: in myoanatomy

Deleted: in tardigrades

established among different arthropod subgroups [52, 53], comparing the muscular systems between the arthropods and onychophorans would be even more difficult. A possible solution for clarifying the homology of individual leg muscles in onychophorans, tardigrades and arthropods would be to identify muscle-specific genes and compare their expression patterns across panarthropods. Recent studies of the NK cluster and NK-linked homeobox genes indeed revealed that at least seven genes (*NK1*, *NK4*, *MSX*, *LBX*, *TLX*, *Nedx* and *Hhex*) might be expressed in the anlagen of specific leg muscles in the onychophoran *E. rowelli* [54, 55]. Clarifying the identity of these muscles in *E. rowelli* and analysing the expression of these genes in tardigrades and arthropods might shed light on their homology among these groups and the evolution of the muscular system in panarthropod legs.

Taken together, our study revealed unprecedented detail of the lobopodial musculature in the onychophoran *E. rowelli*, a member of the Peripatopsidae. Since there is discrepancy between our present and the previous findings (Table 1), the question arises of whether or not this is due to a methodological artefact or represents real interspecific variability. To clarify this question, a member of the second major onychophoran subgroup, the Peripatidae, must first be studied using a similar approach. This would enable a reliable extrapolation across Onychophora as a whole. Likewise, acquiring similar sets of myoanatomical data from various arthropods may help to reconstruct the myoanatomy of lobopods in the last common ancestor of Panarthropoda. Only then it will be possible to infer the lobopodial organisation in early panarthropods by avoiding an onychophoran-biased view.

Ethics. The experiments in this study did not require an approval by an ethical committee. All procedures in this investigation complied with international and institutional guidelines, including the guidelines for animal welfare as laid down by the German Research Foundation (DFG). Animals were collected under the permit numbers SL100159/2011 and SL101720/2016 issued by the NSW National Parks & Wildlife Service and exported under the permit numbers WT2012-8163 and PWS2016-AU-001023 obtained from the Department of Sustainability, Environment, Water, Population and Communities.

Data Accessibility. Supplementary movies showing the 3D reconstructed structures and functional morphology of the lobopods in *E. rowelli* are available within the Dryad Digital Repository: <https://datadryad.org/review?doi=doi:10.5061/dryad.1gh5376> [58]

Authors' Contributions. ISO, AK, HJ and GM designed the study; GM and FP contributed with materials; ISO, MM, HJ and AK acquired the data; ISO, AK, HJ and GM analysed and interpreted the data; ISO, AK and GM drafted the manuscript. All authors revised the drafts and gave final approval for publication.

Competing Interests. The authors declare no competing interests.

Funding. Financial support came from the Conselho Nacional de Desenvolvimento Científico e Tecnológico (CNPq Brazil: 290029/2010-4) and the Zentrale Forschungsförderung of the University of Kassel (ZFF: 1970/2016) to ISO, from the German Research Foundation (DFG: Ma 4147/3-1, 7-1) to GM and from the Deutsches Elektronen-Synchrotron (DESY: I-20170069) to ISO, HJ and GM.

Acknowledgments. We are thankful to members of the Department of Zoology, University of Kassel, for their assistance with animal husbandry. Noel N. Tait is acknowledged for his help with the permits and Dave M. Rowell, Paul Sunnucks, Noel N. Tait, Franziska A. Franke, Sandra Treffkorn and Michael Gerth for their support with collecting the specimens. The authors thank Vladimir Gross (University of Kassel) for constructive comments and critical reading of the manuscript. We also thank Jörg U. Hammel (DESY) for his help with SRμCT. Two anonymous reviewers are cordially acknowledged for providing constructive comments on an early version of the manuscript. The staff of the Department of Sustainability, Environment, Water, Population and Communities of the Australian Government are gratefully acknowledged for providing collecting and export permits.

Formatted: Line spacing: 1.5 lines

Formatted: Font:Italic

References

- Ma, X., Hou, X., Bergström, J. 2009 Morphology of *Luolishania longicrus* (Lower Cambrian, Chengjiang Lagerstätte, SW China) and the phylogenetic relationships within lobopodians. *Arthropod Struct Dev.* **38**, 271–291. (10.1016/j.asd.2009.03.001)
- Rota-Stabelli, O., Kayal, E., Gleeson, D., Daub, J., Boore, J., Telford, M., Pisani, D., Blaxter, M., Lavrov, D. 2010 Ecdysozoan mitogenomics: Evidence for a common origin of the legged invertebrates, the Panarthropoda. *Genome Biol Evol.* **2**, 425–440. (10.1098/rspb.2010.0590)
- Zhang, X., Smith, M. R., Yang, J., Hou, J. B. 2016 Onychophoran-like musculature in a phosphatized Cambrian lobopodian. *Biol Lett.* **12**, 20160492. (10.1098/rsbl.2016.0492)
- Edgecombe, G. D. 2010 Arthropod phylogeny: An overview from the perspectives of morphology, molecular data and the fossil record. *Arthropod Struct Dev.* **39**, 74–87. (10.1016/j.asd.2009.10.002)
- Budd, G. E. 2001 Why are arthropods segmented? *Evol Dev.* **3**, 332–342.
- Budd, G. E. 2003 The Cambrian fossil record and the origin of the phyla. *Integr Comp Biol.* **43**, 157–165.
- Budd, G. E., Telford, M. J. 2009 The origin and evolution of arthropods. *Nature.* **457**, 812–817. (10.1038/nature07890)
- Giribet, G., Edgecombe, G. D. 2019 The phylogeny and evolutionary history of arthropods. *Curr Biol.* **29**, R592–R602. (10.1016/j.cub.2019.04.057)
- Ou, Q., Shu, D., Mayer, G. 2012 Cambrian lobopodians and extant onychophorans provide new insights into early cephalization in Panarthropoda. *Nat Commun.* **3**, 1–7. (10.1038/ncomms2272)
- Ortega-Hernández, J. 2016 Making sense of 'lower' and 'upper' stem-group Euarthropoda, with comments on the strict use of the name Arthropoda von Siebold, 1848. *Biol Rev (Camb).* **91**, 255–273. (10.1111/brv.12168)
- Ou, Q., Mayer, G. 2018 A Cambrian unarmoured lobopodian, †*Lenisambulatrix humboldti* gen. et sp. nov., compared with new material of †*Diania castiformis*. *Sci Rep.* **8**, 13667. (10.1038/s41598-018-31499-y)
- Ortega-Hernández, J. 2015 Lobopodians. *Curr Biol.* **25**, R845–R875. (10.1016/j.cub.2015.07.028)
- Legg, D. A., Sutton, M. D., Edgecombe, G. D., Caron, J. B. 2012 Cambrian bivalved arthropod reveals origin of arthrodization. *Proc R Soc B.* **279**, 4699–4704. (10.1098/rspb.2012.1958)
- Janssen, R., Eriksson, B. J., Budd, G. E., Akam, M., Prpic, N. M. 2010 Gene expression patterns in an onychophoran reveal that regionalization predates limb segmentation in pan-arthropods. *Evol Dev.* **12**, 363–372. (10.1111/j.1525-142X.2010.00423.x)
- Janssen, R., Jörgensen, M., Prpic, N. M., Budd, G. E. 2015 Aspects of dorso-ventral and proximodistal limb patterning in onychophorans. *Evol Dev.* **17**, 21–33. (10.1111/ede.12107)
- Haug, J. T., Mayer, G., Haug, C., Briggs, E. G. 2012 A Carboniferous non-onychophoran lobopodian reveals long-term survival of a Cambrian morphotype. *Curr Biol.* **22**, 1673–1675. (10.1016/j.cub.2012.06.066)
- Budd, G. E. 1998 Arthropod body-plan evolution in the Cambrian with an example from anomalocaridid muscle. *Lethaia.* **31**, 197–210.
- Young, F. J., Vinther, J. 2017 Onychophoran-like myoanatomy of the cambrian gilled lobopodian *Pambdelurion whittingtoni*. *Palaentology.* **60**, 27–54. (10.1111/pala.12269)
- Manton, S. M. 1977 *The Arthropoda: Habits, Functional Morphology, and Evolution*. Oxford, England: Clarendon Press
- Schmidt-Rhaesa, A. 2001 Tardigrades - Are they really miniaturized dwarfs? *Zool Anz.* **240**, 549–555.
- Gross, V., Treffkorn, S., Reichelt, J., Epple, L., Lüter, C., Mayer, G. 2019 Miniaturization of tardigrades (water bears): Morphological and genomic perspectives. *Arthropod Struct Dev.* **48**, 12–19. (10.1016/j.asd.2018.11.006)
- Gross, V., Müller, M., Hehn, L., Ferstl, S., Allner, S., Dierolf, M., Achterhold, K., Mayer, G., Pfeiffer, F. 2019 X-ray imaging of a water bear offers a new look at tardigrade internal anatomy. *Zoological Lett.* **5**, 14. (10.1186/s40851-019-0130-6)
- Hou, X. G., Bergström, J. 1995 Cambrian lobopodians - ancestors of extant onychophorans? *Zool J Linn Soc.* **114**, 3–19.
- Hoyle, G., Williams, M. 1980 The musculature of *Peripatus* and its innervation. *Phil Trans R Soc B.* **288**, 481–510.
- Manton, S. M. 1950 The evolution of arthropodan locomotory mechanisms. - Part I. The locomotion of *Peripatus*. *J Linn Soc Lond Zool.* **41**, 529–570.
- Manton, S. M. 1953 Locomotory habits and the evolution of the larger arthropod groups. In *Evolution [Symposia of the Society for Experimental Biology, vol. 7]*. (ed. A. eds. pp. 339–376. London: Society for Experimental Biology.
- Manton, S. M. 1973 The evolution of arthropodan locomotory mechanisms. Part 11: Habits, morphology and evolution of the Uniramia (Onychophora, Myriapoda and Hexapoda) and comparisons with the Arachnida, together with a functional review of uniramian musculature. *Zool J Linn Soc.* **53**, 257–375.
- Snodgrass, R. E. 1938 Evolution of the Annelida, Onychophora and Arthropoda. *Smith Misc Coll.* **97**, 1–159.
- Maas, A., Waloszek, D., Haug, J. T., Müller, K. J. 2007 A possible larval roundworm from the Cambrian 'Orsten' and its bearing on the phylogeny of Cycloneuralia. *AAP Memoir.* **34**, 499–519.
- Ou, Q., Liu, J., Shu, D., Han, J., Zhang, Z., Wan, X., Lei, Q. 2011 A rare onychophoran-like lobopodian from the lower Cambrian Chengjiang Lagerstätte, southwestern China, and its phylogenetic implications *J Paleontol.* **85**, 587–594. (10.1666/09-147R2.1)
- Birket-Smith, S. J. R. 1974 The anatomy of the body wall of Onychophora. *Zool Jahrb Abt Anat Ontog Tiere.* **93**, 123–154.
- Gaffron, E. 1885 Beiträge zur Anatomie und Histologie von *Peripatus*. *Zool Beitr.* **1**, 33–60.
- Pflugfelder, O. 1968 Onychophora. In *Grosses Zoologisches Praktikum*. (ed. A. eds. G. Cizhak), pp. 1–42. Stuttgart: Gustav Fischer.
- Lankester, E. R. 1904 The structure and classification of the Arthropoda. *Q J Microsc Sci.* **47**, 523–582.
- Müller, M., Oliveira, I. S., Allner, S., Ferstl, S., Bidola, P., Mechlem, K., Fehringer, A., Hehn, L., Dierolf, M., Achterhold, K., et al. 2017 Myoanatomy of the velvet worm leg revealed by laboratory-based nanofocus X-ray source tomography. *Proc Natl Acad Sci USA.* **114**, 12378–12383. (10.1073/pnas.1710742114)
- Oliveira, I. S., Bai, M., Jahn, H., Gross, V., Martin, C., Hammel, J. U., Zhang, W., Mayer, G. 2016 Earliest onychophoran in amber reveals Gondwanan migration patterns. *Curr Biol.* **26**, 2594–2601. (10.1016/j.cub.2016.07.023)
- Jahn, H., Oliveira, I. S., Gross, V., Martin, C., Hipp, A., Mayer, G., Hammel, J. U. 2018 Evaluation of contrasting techniques for X-ray imaging of velvet worms (Onychophora). *Journal of Microscopy.* **270**, 343–358. (10.1111/jmi.12688)
- Baer, A., Mayer, G. 2012 Comparative anatomy of slime glands in Onychophora (velvet worms). *J Morphol.* **273**, 1079–1088. (10.1002/jmor.20044)
- Schindelin, J., Arganda-Carreras, I., Frise, E., Kaynig, V., Longair, M., Pietzsch, T., Preibisch, S., Rueden, C., Saalfeld, S., Schmid, B., et al. 2012 Fiji: an open-source platform for biological-image analysis. *Nat Meth.* **9**, 676–682. (10.1038/nmeth.2019)
- Oliveira, I. S., Mayer, G. 2017 A new giant egg-laying onychophoran (Peripatopsidae) reveals evolutionary and biogeographical aspects of Australian velvet worms. *Org Divers Evol.* **17**, 375–391. (10.1007/s13127-016-0321-3)
- Oliveira, I. S., Ruhberg, H., Rowell, M. D., Mayer, G. 2018 Revision of Tasmanian viviparous velvet worms (Onychophora: Peripatopsidae) with descriptions of two new species. *Invertebr Syst.* **32**, 909–932. (doi.org/10.1071/IS17096)
- Martin, C., Gross, V., Hering, L., Tepper, B., Jahn, H., Oliveira, I. S., Stevenson, P. A., Mayer, G. 2017 The nervous and visual systems of onychophorans and tardigrades: learning about arthropod evolution from their closest relatives. *J Comp Physiol A.* **203**, 565–590. (10.1007/s00359-017-1186-4)
- Manton, S. M. 1967 The polychaete *Spinther* and the origin of the Arthropoda. *J Nat Hist.* **1**, 1–22.
- Oliveira, I. S., Mayer, G. 2013 Apodemes associated with limbs support serial homology of claws and jaws in Onychophora (velvet worms). *J Morphol.* **274**, 1180–1190. (10.1002/jmor.20171)
- Mayer, G., Franke, F. A., Treffkorn, S., Gross, V., Oliveira, I. S. 2015 Onychophora. In *Evolutionary Developmental Biology of Invertebrates*. (ed. A. eds. A. Wanninger), pp. 53–98. Berlin: Springer.
- Mayer, G., Oliveira, I. S., Baer, A., Hammel, J. U., Gallant, J., Hochberg, R. 2015 Capture of prey, feeding, and functional anatomy of the jaws in velvet worms (Onychophora). *Integr Comp Biol.* **55**, 217–227. (10.1093/icb/004)
- Martin, C., Gross, V., Pflüger, H.-J., Stevenson, P. A., Mayer, G. 2017 Assessing segmental versus non-segmental features in the ventral nervous system of onychophorans (velvet worms). *BMC Evol Biol.* **17**, 3. (10.1186/s12862-016-0853-3)
- Smith, M. R., Ortega-Hernández, J. 2014 *Hallucigenia*'s onychophoran-like claws and the case for Tactopoda. *Nature.* **514**, 363–366. (10.1038/nature13576)
- Gross, V., Mayer, G. 2019 Cellular morphology of leg musculature in the water bear *Hypsibius exemplaris* (Tardigrada) unravels serial homologies. *Royal Society Open Science.* **this issue**.
- Bicknell, R. D. C., Klinkhamer, A. J., Flavel, R. J., Wroe, S., Peterson, J. R. 2018 A 3D anatomical atlas of appendage musculature in the chelicerate arthropod *Limulus polyphemus*. *PLOS*

51. Grams, M., Wirkner, C., Runge, J. 2018 Serial and special: Comparison of podomeres and muscles in tactile vs walking legs of whip scorpions (Arachnida, Uropygi). *Zool Anz.* **273**, 75–101. (10.1016/j.jcz.2017.06.001)

52. Boxhall, G. 1997 Comparative limb morphology in major crustacean groups: the coxa-basis joint in postmandibular limbs. In *Arthropod Relationships*. (ed. Aeds). pp. 155–167. London: Chapman & Hall.

53. Boxhall, G. A. 2004 The evolution of arthropod limbs. *Biol Rev.* **79**, 253–300.

54. Treffkorn, S., Kahnke, L., Hering, L., Mayer, G. 2018 Expression of NK cluster genes in the onychophoran *Euperipatoides rowelli*: implications for the evolution of NK family genes in nephrozoans. *EvoDe.* **9**, 17. (10.1186/s13227-018-0105-2)

55. Treffkorn, S., Mayer, G. 2019 Expression of NK genes that are not part of the NK cluster in the onychophoran *Euperipatoides rowelli* (Peripatopsidae). *BMC Dev Biol.* **19**, 7. (10.1186/s12861-019-0185-9)

56. Snodgrass, R. E. 1958 Evolution of arthropod mechanisms. *Smithson Institution Miscellaneous Collections.* **138**, 1–77.

57. Oliveira, I. S., Wieloch, A. H., Mayer, G. 2010 Revised taxonomy and redescription of two species of the Peripatidae (Onychophora) from Brazil: a step towards consistent terminology of morphological characters. *Zootaxa.* **2493**, 16–34.

58. Oliveira, I. S., Kumerics, A., Jahn, H., Müller, M., Pfeiffer, F. & Mayer, G. 2019 Data from: Functional morphology of a lobopod: Case study of an onychophoran leg. Dryad Digital Repository. (<https://datadryad.org/review?doi=doi:10.5061/dryad.1qh5376>)

Deleted:

- Formatted: English (UK)
- Formatted: English (UK)
- Formatted: English (UK)
- Formatted: English (UK)
- Formatted: English (UK)
- Formatted: English (UK)
- Formatted: English (UK)
- Formatted: English (UK)
- Deleted:

Figure and Table captions

Figure 1. External and internal anatomy of the lobopod in *E. rowelli*. Volume rendering based on nanoCT data from left mid-trunk leg in posterior (a,b,d,f) and anterior views (c,e). Dorsal is up in all images. Arrowheads demarcate position of nephridial opening; arrows indicate spineless integumentary fold located between second and third spinous pads. (a) External structures. (b) Internal non-muscular structures. (c–f) Myoanatomy. Individual muscles are highlighted in different colours and numbered as in main text (summarised in Table 1). Note that e,f illustrate same perspective as c,d except that some outer muscles (#3, #5 in e, and #1, #6, #9, #10; #13 in f) were excluded to better visualise internal muscles. Abbreviations: ap, foot apodeme; br, foot bridge; cf, large collagen fibres; cl, claw; dp, dermal papilla; he, haemocyte; ln, leg nerves; mc, median commissure; nb, nephridial bladder; nc, nerve cord; nd, nephridial duct; ne, nephrocyte; rc, ring commissure; sa, sacculus; sp, spinous pads.

Figure 2. External and internal anatomy of the foot in *E. rowelli*. Scanning electron micrographs (a,b) and volume rendering of nanoCT data (c). Dorsal is up in all images. Arrows in a,b indicate sensory bristles of distal foot papillae. (a) Foot with protracted claws. Note everted dorsal sac (pseudocoloured turquoise). (b) Sagittal section of foot with retracted claws. Note contracted eversible dorsal sac and distal foot papilla bent distally. Also note attachment of claw retractor muscle (pseudocoloured yellow) at claw base and presence of haemocytes (pseudocoloured pink) in foot cavity. (c) Virtual sagittal section of foot. Leg and foot cavities illustrated in light-brown/orange; muscles colour-coded as in figure 1 and numbered as in main text (summarised in Table 1). Arrowheads point to haemocytes located in foot cavity. Abbreviations: ap, foot apodeme; cl, claw; es, eversible dorsal sac; fc, foot cavity; fp, distal foot papilla. Scale bars: 30 μm in (a) and 50 μm in (b).

Figure 3. Lacunar system of the lobopod in *E. rowelli*. Volume rendering of nanoCT data from left mid-trunk leg showing space occupied by haemolymph. Dorsal is up in a,b,d–f; posterior is up in c. Compartments of leg cavity are numbered I–IV. (a,b) Overview of lacunar system in leg and foot. Note that transverse, haemal ring channels correspond to external rings of dermal papillae (semi-transparent in a). (c–e) Horizontal (c), cross (d), and sagittal (e) virtual sections of leg. Arrow in e points to narrow channel connecting cavities of leg and trunk. Muscles are numbered as in main text (summarised in Table 1). Note that anteroposterior and dorsoventral septal muscles (#10, #11) subdivide leg cavity into four compartments (I–IV in d), with two dorsal compartments (Ist and IInd) being largest. (f) Same virtual section and perspective as in d showing arrangement of internal structures to the exclusion of leg cavity. Abbreviations: ap, foot apodeme; br, foot bridge; cf, large collagen fibres; cl, claw; fc, foot cavity; ln, leg nerve; mc, median commissure; nb, nephridial bladder; nc, nerve cord; nd, nephridial duct; ne, nephrocytes; no, nephridial opening; sa, sacculus; rc, ring commissure; tc, transverse haemal ring channels; tr, trunk.

Deleted ;

Figure 4. Overview of individual muscles associated with the lobopod of *E. rowelli*. Volume rendering of nanoCT data from left mid-trunk leg illustrating shape and position of all fifteen lobopodial muscles. Dorsal is up in all images. Muscles are numbered as in main text (summarised in Table 1). Body surface is semi-transparent; “extrinsic” portions of muscles #1–#4, and #8 are not shown.

Figure 5. Lobopodial muscles extending into the trunk of *E. rowelli*. Volume rendering based on SR μ CT data (a,b,d) and scanning electron micrograph (c). Muscles are numbered as in main text (summarised in Table 1). (a) Virtual slice of mid-body trunk segment. (b) Internal perspective of “extrinsic” leg muscles viewed from trunk. Note median position of anteroproximal (#7) and posteroproximal leg muscles (#8) crossing the main channel (asterisks) linking cavities of leg and trunk. (c) Imprints of ventral and pre-ventral organs on inner surface of moulted cuticle. (d) Ventral view of lobopod showing relationship of leg depressor (#2) and leg remotor (#4) with ventral and pre-ventral organs. Abbreviations: gu, gut; le, leg; nc, nerve cord; pv, pre-ventral organ; tm, transverse musculature; vo, ventral organ. Scale bar: 50 μm in (c).

Figure 6. Lobopod movements during locomotion in *E. rowelli*. Photograph (top) and selected frames extracted from high-speed camera footage (500 fps) of walking specimen (a–h) with corresponding diagrams (a'–h') showing major movements of individual lobopod. Asterisks in top image highlight segments with synchronously moving lobopods. Gradient colour bar indicates continuous transition between major steps in locomotion. Arrowheads (in e–g) point to everted dorsal sac. Note complete retraction of claws at the end of levation (b) and fully protracted claws in the middle of depression (f).

Table 1. Onychophoran leg muscles revealed in the present and previous comprehensive studies on myoanatomy. Asterisks indicate muscles with uncertain identity.

Muscle	present study Euperipatoides rowelli (Peripatopsidae)	Snodgrass [28] Peripatoides novaecelandiae (Peripatopsidae)	Pflugfelder [33] Paraperipatus amboinensis (Peripatopsidae)	Manton [27, 43] Peripatopsis spp. (Peripatopsidae)	Birket-Smith [31] Peripatus dominicae (Peripatidae)	Hoyle & Williams [24] Peripatus dominicae (Peripatidae)	Comments
#1	leg levator	dorsal remotor of leg		dorsal remotor of the leg	dorsal remotor of the lobopod	remotor	
#2	leg depressor	ventral promotor of leg	(1) oblique and (2) transverse trunk musculature	ventral promotor of the leg	ventral promotor of the lobopod	anterior depressor	Pflugfelder [33], followed by Manton [27, 43], recognised the muscles #1–#4 based on their extrinsic parts, whereas their intrinsic parts were described together as a "thick peripheral layer of intrinsic leg fibres" (= "peripheral muscles of basal part of leg" by Snodgrass [56]; "superficial longitudinal muscle" by Manton [27, 43]).
#3	leg promotor	dorsal promotor of leg		dorsal promotor of the leg	dorsal promotor of the lobopod	promotor	
#4	leg remotor	ventral remotor of leg	ventral remotor of the leg	ventral remotor of the lobopod	posterior depressor		
#5	anterior leg rotator	-	-	-	-	-	
#6	posterior leg rotator	-	-	-	-	-	
#7	anteroproximal leg muscle	transverse muscle of leg base	-	transverse muscle of leg base	anterior retractor of the nephropore	-	
#8	posteroproximal leg muscle	-	-	-	posterior retractor of the nephropore	-	
#9	leg flexor	-	-	-	-	-	
#10	anteroposterior septal muscle	anteroposterior septal muscle	-	transverse muscle	anteroposterior lobopod muscle sheet	septal muscle	
#11	dorsoventral septal muscle	-	-	-	transverse lobopod muscle sheet	-	The dorsoventral septal muscle (#11) was identified as muscle "14" by Birket-Smith [31] in reference to Snodgrass [28], although the latter used "14" for a muscle that corresponds to the anteroproximal leg muscle (#7) in our study instead.
#12	claw retractor	two-branched retractor of claws	levator of claws	levator muscle	lifter of the claws	retractor of the claw	The "two branches" of the claw retractor described by Snodgrass [28], namely muscles "19a" and "19b", correspond to the single muscle #12 in the present study.
#13	foot depressor	flexor of distal leg rings	flexor muscle of penultimate foot segment	depressor muscle	flexor (protruder) muscle of the foot pads	-	This muscle was recently identified as "foot retractor muscle" by Oliveira & Mayer [44] and as a "protractor" of the foot by Müller et al. [35].
#14	bridge constrictor	-	-	-	-	-	
#15	foot constrictor	circular muscles of foot	-	-	protractor muscle of the claws	small circular fibres (...) of the foot	
*	-	flexor muscle of leg	-	-	flexor muscle of the lobopod	-	This muscle possibly represents a subset of muscle fibres belonging to the leg promotor (#3) of the present study.

*	-	-	-	(superficial) oblique muscles	-	-	
*	-	-	(1) adductor and (2) remotor of the claws	-	depressor of the claws	-	
*	-	-	-	circular muscles in the limb wall	external circular muscle	(...) circular muscle of the body wall running into the leg	
*	-	-	-	14a	-	-	Manton [27, 43] neither named nor further characterized these muscles.
*	-	-	-	14b	-	-	
*	-	-	-	-	retractor muscle of the lobopod	-	This muscle possibly represents a subset of muscle fibres belonging to the leg levator (#1) in the present study.
*	-	-	-	-	nephropore protractor muscle	-	
*	-	-	-	-	ribbon-shaped lobopod muscle	-	
*	-	-	-	-	tensor muscle of the horizontal septum	-	
*	-	-	-	-	internal, caudal sphincter muscle of the lobopod	-	
*	-	-	-	-	retractor muscle of the eversible sac	-	Birket-Smith [31] used the term "eversible sac" for the so-called coxal vesicle, which occurs on the ventral surface of the leg (see, e.g., [57]), rather than for the eversible dorsal sac associated with the foot (sensu [44]).
*	-	-	-	-	frontal sphincter muscle of the lobopod	-	
*	-	-	-	-	external, caudal sphincter muscle of the lobopod	-	
*	-	-	-	-	-	levator	
Number of identified muscles	15	10	6	12	22	9	

Table 2. Relative volumes of structures within the leg of *E. rowelli*. For volume estimates of individual muscles, only their “intrinsic” portions were taken into account.

Structure, cavity, cell type or tissue	Total number of voxels	Relative volume (%)
Entire leg	178,999,899	100.00
Leg cavity	81,829,369	45.71
Muscles		
#1 leg levator	4,153,183	2.32
#2 leg depressor	3,933,669	2.20
#3 leg promotor	7,450,512	4.16
#4 leg remotor	2,072,857	1.16
#5 anterior leg rotator	256,237	0.14
#6 posterior leg rotator	284,769	0.16
#7 anteroproximal leg muscle	259,261	0.14
#8 posteroproximal leg muscle	210,226	0.12
#9 leg flexor	916,491	0.51
#10 anteroposterior septal muscle	3,059,215	1.71
#11 dorsoventral septal muscle	1,811,738	1.01
#12 claw retractor	1,497,049	0.84
#13 foot depressor	266,166	0.15
#14 bridge constrictor	59,578	0.03
#15 foot constrictor	227,010	0.13
Total		14.78
Non-muscular structures		
Nervous system	970,481	0.54
Excretory system	10,676,174	5.96
Haemo- and nephrocytes	1,956,540	1.09
Large collagen fibres	81,718	0.05
Claws	958,206	0.54
Foot apodeme	117,422	0.07
Cuticle	10,365,894	5.79
Other cells and tissues	45,586,134	25.47
Total		39.51

*Structures not segmented herein, including epidermal cells, peripheral sensory neurons, extracellular matrix, and connective tissue.

Supplementary Figure 1. Spinous pads of the lobopod in *E. rowelli*. Scanning electron micrograph (a) and volume rendering based on nanoCT data (b) from left mid-trunk leg in ventrodorsal view. Distal is up in both images. Spinous pads are numbered. Note the position of the spineless integumentary fold between the second and third spinous pads. Abbreviation: if, spineless integumentary fold.

Supplementary Figure 2. Musculature and haemolymph system of the foot in *E. rowelli*. Volume rendering based on nanoCT data from left mid-trunk leg. Dorsal is up in both images. Body surface is semi-transparent. (a) Detail of the haemolymph channel within the foot bridge connecting the leg to the foot. The foot constrictor (#15, as in the main text) may regulate the hydrostatic pressure inside the foot. (b) Distribution of haemo- and nephrocytes within the leg and foot. Note that nephrocytes are larger and restricted to the leg cavity. Abbreviations: ap, foot apodeme; br, foot bridge; cl, claw; fc, foot cavity; he, haemocyte; lc, leg cavity; ne, nephrocyte.

Supplementary Figure 3. Elements of the nervous and excretory system in the lobopod of *E. rowelli*. Volume rendering based on nanoCT data from left mid-trunk leg. Dorsal is up in all images. Body surface is semi-transparent. Abbreviations: ap, foot apodeme; cl, claw; ln, leg nerves; mc, median commissure; nb, nephridial bladder; nc, nerve cord; nd, nephridial duct; no, nephridial opening; rc, ring commissure; sa, sacculus.

Supplementary Figure 4. Internal view of a moulted skin of *E. rowelli*. Scanning electron micrograph. Dorsal is up. Note the lack of apodeme-like structures on the dorsolateral body wall. Abbreviations: dm, dorsomedian furrow; dp, dermal papilla; le, leg.

Supplementary Figure 5. Internal organisation of the lobopod in *E. rowelli*. Volume rendering based on nanoCT data from left mid-trunk leg; virtual section along the proximodistal lobopod axis. Dorsal is up in both images. (a) Leg in posterior view. (b) Leg in anterior view. Individual muscles are highlighted in

different colours and numbered as in main text (summarised in Table 1). Abbreviations: cl, claw; if, spineless integumentary fold; nb, nephridial bladder; ne, nephrocytes; sp, spinous pad.

Supplementary Figure 6. Spatial relationship between proximal leg muscles and the excretory organ in *E. rowelli*. Volume rendering based on nanoCT data from left mid-trunk leg illustrating the anteroproximal (#7) and posteroproximal (#8) leg muscles and the nephridial system of the leg (in brown). Dorsal is up in all images except the centre image in **a** (distal is up). Body surface is semi-transparent. Note that leg muscles are not associated with the nephridial opening. **(a)** Lobopod viewed from different perspectives. **(b)** Detail of the leg viewed from trunk. Abbreviations: ap, foot apodeme; cl, claw; nb, nephridial bladder; nd, nephridial duct; no, nephridial opening; sa, sacculus.

Supplementary Figure 7. Morphology of the claw retractor muscle in *E. rowelli*. Volume rendering based on nanoCT data from left mid-trunk leg. Dorsal is up in all images. Body surface is semi-transparent in **a**. **(a)** Structure and position of the claw retractor muscle (#12) inside the leg. Arrowheads point to set of muscle fibres that branch off ventrally from this muscle. **(b)** Spatial relationship between the claw retractor (#12) and the leg promotor (#3). Note that the dorsoproximal fibres of the claw retractor (arrow) fuses with those of the leg promotor. Abbreviations: ap, foot apodeme; cl, claw; tr, trunk.

Supplementary Figure 8. Peripheral arrangement of leg muscles in *E. rowelli*. Volume rendering based on SR μ CT data from left mid-trunk leg illustrating the lobopod in virtual cross section (top left image). Dorsal is up in all images. Body surface is semi-transparent (bottom image). Individual muscles are highlighted in different colours and numbered as in main text (summarised in Table 1). Abbreviation: lg, leg.

Formatted: Line spacing: multiple 1.15 li, Tabs: 9.11 cm, Left

Deleted: ... [3]

Although only limited information is available on the myoanatomy of fossil lobopodians, these animals might have possessed a well-developed leg musculature with “extrinsic” portions [3, 30], which might correspond to the four most prominent leg muscles (#1–#4) of *E. rowelli*. Even the fan-shaped nature of the individual muscles might have existed in the extinct, marine lobopodians [3, 18], like in extant, terrestrial onychophorans, probably due to the soft body surface of these animals, which otherwise would be deformed during muscle contraction.

T

Supplementary Movie 1. External and internal anatomy of the lobopod in *E. rowelli*.

Supplementary Movie 2. Haemal space inside the lobopod of *E. rowelli*.

Supplementary Movie 3. Muscles involved in hydrostatic pressure control, stabilisation and claw movements in *E. rowelli*.

Supplementary Movie 4. Individual muscles composing the lobopod of *E. rowelli*.

Supplementary Movie 5. High-speed footage of a walking specimen of *E. rowelli*.